# A nonparametric mixed exponentially weighted moving average-moving average control chart with an application to gas turbines

**Muhammad Ali Raza**[1], **Farah Tariq**[1], **Abdullah A. Zaagan**[2], **Gideon Mensah Engmann**[3], **Ali M. Mahnashi**[2], **Mutum Zico Meetei**[2]*

**1** Department of Statistics, Government College University Faisalabad, Faisalabad, Pakistan, **2** Department of Mathematics, College of Science, Jazan University, Jazan, Kingdom of Saudi Arabia, **3** Department of Biometry, C. K. Tedam University of Technology and Applied Sciences, Navrongo, Ghana

* mmeetei@jazanu.edu.sa

**Data Availability Statement:** All relevant data are within the paper.

## Abstract

This study aims to develop a nonparametric mixed exponentially weighted moving average-moving average (NPEWMA-MA) sign control chart for monitoring shifts in process location, particularly when the distribution of a critical quality characteristic is either unknown or non-normal. In literature, the variance expression of the mixed exponentially weighted moving average-moving average (EWMA-MA) statistic is calculated by allowing sequential moving averages to be independent, and thus the exclusion of covariance terms results in an inaccurate variance expression. Furthermore, the effectiveness of the EWMA-MA control chart deteriorates when the distribution of a critical quality characteristic deviates from normality. The proposed NPEWMA-MA sign control chart addresses these by utilizing the corrected variance of the EWMA-MA statistic and incorporating the nonparametric sign test into the EWMA-MA charting structure. The chart integrates the moving average (MA) statistic into the exponentially weighted moving average (EWMA) statistic. The EWMA-MA charting statistic assigns more weight to recent w samples, with weights for previous observations decling exponentially. Monte Carlo simulations assess the chart's performance using various run length (RL) characteristics such as average run length (ARL), standard deviation of run length (SDRL), and median run length (MRL). Additional measures for overall performance include the average extra quadratic loss (AEQL) and relative mean index (RMI). The proposed NPEWMA-MA sign control chart demonstrates superior performance compared to existing nonparametric control charts across different symmetrical and asymmetric distributions. It efficiently detects process shifts, as validated through both a simulated study and a real-life example from a combined cycle power plant.

## 1. Introduction

Quality control encompasses a range of procedures aimed at upholding and enhancing product quality in accordance with predefined benchmarks. Its primary objective is to ensure the

**Funding:** This research was funded by the Deputyship for Research and Innovation, Ministry of Education in Saudi Arabia, through the project number ISP-2024 (to M.Z.M.). The funders had no role in study design, data collection and analysis, decision to publish, or preparation of the manuscript.

**Competing interests:** The authors have declared that no competing interests exist.

consistency of product manufacturing and its alignment with clients' requirements. Statistical Process Control (SPC) is a combination of diverse statistical methods employed to enhance the quality of production processes or services. Among these techniques, the control chart stands out as a particularly valuable tool within SPC, initially presented by Walter A. Shewhart in the 1920s [1]. While these control charts are readily implementable and adept at detecting significant process shifts, they face limitations in detecting minor shifts due to their reliance solely on current sample data. To address this limitation, memory-type control charts have emerged in the literature. Examples include the Cumulative Sum (CUSUM) control chart proposed by Page [2], the EWMA control chart introduced by Roberts [3], and the MA control chart discussed by Roberts [4]. These advanced charts excel at swiftly identifying subtle to moderate alterations in process parameters by capitalizing on information from both the current and preceding samples.

To enhance the performance of control charts, many researchers suggested combined control charting techniques. For instance, Lucas [5] introduced a combined Shewhart-CUSUM control chart using the features of the Shewhart chart to detect larger shifts and CUSUM in identifying smaller changes. Similarly, Lucas and Saccucci [6] suggested a combined Shewhart-EWMA control chart to detect small to large process shifts efficiently. Moreover, Shamma et al. [7] designed the double exponentially weighted moving average (DEWMA) control chart by incorporating two EWMA statistics, which was discussed later by many authors, for example, Zhang and Chen [8], Mahmoud and Woodall [9], and Haq [10]. Additionally, Abbas et al. [11] proposed a mixed EWMA-CUSUM control chart by integrating EWMA statistics into CUSUM charting statistics to detect small shifts rapidly. Zaman et al. [12] developed a similar technique by combining CUSUM statistics with the EWMA control chart for rapidly identifying minor variations in process location. By combining two MA statistics, Khoo and Wong [13] introduced a double moving average (DMA) control chart; however, the variance of the DMA statistics presented in this paper is inaccurate. Later, Alevizakos et al. [14] suggested the corrected version of the DMA control chart. A triple exponentially weighted moving average (TEWMA) control chart was developed by Alevizakos et al. [15] for rapidly recognizing small variations in the process mean. Integrating one memory-type charting statistic into the other (or the similar) gives past data more weight than current observation or statistic. Although the zero-state out-of-control (OOC) RL performance of these mixed memory-type control charts (MMTCC) is better than the conventional memory-type control charts, the steady-state OOC RL performance deteriorates. These MMTCC allocate more weights to older observations compared to the current one. For more details about the consequences of the MMTCC, see Knoth et al. [16].

All the aforementioned control charts assume that a certain quality characteristic is normally distributed. But, in many real-life situations, there is a dearth of information to justify this assumption which can influence the competence of the control charts [17]. In such circumstances, distribution-free or nonparametric control charts can be used alternatively to monitor the process parameters that do not require any distributional assumptions. Chakraborti et al. [18] provided an updated overview of the univariate and multivariate nonparametric control charts, and pointed out the simplicity, robustness, and efficiency of the nonparametric control charts. The in-control (IC) RL of nonparametric control charts is constant for continuous distributions. For details, see Chakraborti et al. [19].

In the SPC literature, several distribution-free control charts (DFCC) have been proposed for monitoring process variability. For instance, Bakir and Reynolds [20] proposed a DFCC for efficient process location monitoring using the Wilcoxon signed-rank test. To monitor process location, Amin and Searcy [21] proposed a distribution-free EWMA signed-rank chart. Amin et al. [22] presented the nonparametric Shewhart and CUSUM sign control charts

for monitoring the process median. Bakir [23] introduced a signed-rank statistic-based She-whart-type control chart which is robust to non-normality and the presence of outliers. More-over, Bakir [24] used a signed-rank-like statistic to develop a DFCC for monitoring the process mean when the IC process mean is unknown. A nonparametric control chart based on the change-point model was developed by Hawkins and Deng [25] to identify minor to moder-ate changes in the process mean. Yang et al. [26] proposed the nonparametric EWMA sign control chart for detecting small changes in the process mean. Graham et al. [27] explored a Phase-II nonparametric EWMA control chart based on the signed-rank statistic. Mukherjee et al. [28] suggested a nonparametric CUSUM control chart based on the exceedance statistics for identifying a shift in the process location for a continuous distribution. Lu [29] proposed an extended nonparametric EWMA sign control chart to improve its performance in detecting minor shifts in the process. Pawar et al. [30] suggested sign and signed-rank statistics based distribution-free moving average control charts to identify changes in the process location. Raza et al. [31] developed distribution-free homogeneously weighted moving average control charts using sign and signed-rank statistics to monitor the shift in process location. Li [32] pro-posed an adaptive CUSUM control chart for detecting arbitrary distributional changes. Abbas et al. [33] developed a nonparametric progressive control chart based on the Wilcoxon signed-rank statistic to identify changes in process location. Raza et al. [34] presented a distribution-free DEWMA control chart that utilizes signed ranks to identify changes in the location. Shaf-qat et al. [35] presented a nonparametric modified arcsine EWMA control based on repetitive sampling to detect the small changes in process location. To monitor mixed continuous and categorical data, Li [36] proposed a nonparametric adaptive EWMA control chart using a self-starting technique. For more details, see Haq [37], Raza et al. [38], Aslam et al. [39], Salamai [40], Al-Omari et al. [41], Haridy et al. [42], Raza et al. [43, 44], and Triantafyllou [45].

In recent literature, Sukparungsee et al. [46] designed a mixed EWMA-MA control chart for normally distributed data to identify small to large process shifts efficiently. The variance term of the EWMA-MA statistic provided in this paper is imprecise because the covariance terms among the MAs are omitted by considering the succeeding moving averages as indepen-dent. Recently, Raza et al. [47] presented a corrected version of a mixed EWMA-MA control chart to address the previously identified issues with the variance expression in the presence of a normal distribution. The robustness analysis shows that the EWMA-MA control chart is very resilient to deviations from normality, particularly for smaller values of $\lambda$. Nevertheless, the effectiveness of the control chart declines as the values of $\lambda$ increase. To address this issue, we propose a NPEWMA-MA sign control chart that monitors a range of shifts in the location parameter using the corrected variance of the EWMA-MA statistic.

The paper is organized as follows: Section 2 describes the NPEWMA-MA control chart design, both with and without arcsine transformation. In Section 3, the performance of the proposed chart is assessed under various continuous distributions. Section 4 covers a detailed RL profile comparison of the proposed and existing nonparametric control charts. A real-life application of the proposal is provided in Section 5. Finally, the paper concludes in Section 6.

## 2. The NPEWMA-MA control chart

The suggested NPEWMA-MA control chart is constructed using the sign test. This test was introduced by Arbuthnott [48]. This test is distribution-free and is based on the plus and minus signs. It is used to test the hypothesis that the probability of plus signs (+) is equal to the probability of minus (−) signs equivalently testing the hypothesis that the median of an under-lying process distribution is equal to a specified value. The process is IC when the probability of the plus sign is equal to the probability of the minus sign, i.e., $p_{\text{plus sign}} = p_{\text{minus sign}} = p_0 =$

0.50. The deviation from $p_0$ = 0.50, i.e., $\Delta = |p_0−p_1|$ indicates the OOC process state. The design structure of the NPEWMA-MA control chart is as follows:

## 2.1. The EWMA-MA sign control chart

Suppose $X$ represents a certain quality characteristic with a target or median value equal to $\theta$. Let $Y$ denotes the deviation of $X$ from the target value $\theta$ with probability $p = P(Y>0)$. If $p = p_0$ = 0.5, the process is said to be IC, otherwise, the process is OOC, i.e., $p = p_1 \neq 0.5$. Let the quality characteristic $X_{ij}$, $i = 1,2,\ldots,m$ and $j = 1,2,\ldots n$ is distributed independently and identically taken from $X$ to examine the variation from the process target value $\theta$. Then define:

$$Y_{ij} = X_{ij} - \theta \text{ and } I_{ij} = \begin{cases} 1 & for \ Y_{ij} > 0 \\ 0 & otherwise \end{cases} \tag{1}$$

Let $S_i$ be the number of positive signs, i.e., $S_i = \sum_{j=1}^{n} I_{ij}$ which follows a binomial distribution with parameters $\left(n, \frac{1}{2}\right)$ for an IC process. Then, the moving average statistic $MA_i$ of span $w$ at time $i$ is defined as:

$$MA_i = \begin{cases} \dfrac{\sum_{l=1}^{i} S_l}{i} & \text{for } i < w \\ \dfrac{\sum_{l=i-w+1}^{i} S_l}{w} & \text{for } i \geq w \end{cases} \tag{2}$$

The MA statistic is integrated into the EWMA statistic to develop the NPEWMA-MA sign statistic. The plotting statistic of the NPEWMA-MA sign chart is:

$$Z_{S_i} = \eta MA_i + (1 - \eta)Z_{S_{i-1}} \qquad i = 1, 2, 3, \ldots \tag{3}$$

where $\eta$ ($0<\eta \leq 1$) is the smoothing parameter and the preliminary value of $Z_{S_i}$ is equal to the mean value of $S$, i.e., $Z_{S_0} = \mu_0 = np_0$. The statistic $Z_{S_i}$ can also be written as:

$$Z_{S_i} = \eta \sum_{j=0}^{i-1} (1 - \eta)^j MA_{i-j} + (1 - \eta)^i Z_{S_0} \tag{4}$$

The expectation of the plotting statistic $Z_{S_i}$ is:

$$E(Z_{S_i}) = \eta \sum_{j=0}^{i-1} (1 - \eta)^j E(MA_{i-j}) + (1 - \eta)^i E(Z_{S_0})$$

$$E(Z_{S_i}) = \mu_0 = np_0 \tag{5}$$

The variance of the statistic $Z_{S_i}$ is:

$$Var(Z_{S_i}) = \eta^2 \sum_{j=0}^{i-1} (1 - \eta)^{2j} var(MA_{i-j})$$
$$+ 2\eta^2 \sum_{k_1=1}^{i-1} \sum_{k_2=k_1+1}^{i} (1 - \eta)^{2i-k_1-k_2} Cov(MA_{k_1}, MA_{k_2}) \tag{6}$$

where the variance of *MA* is:

$$Var(MA_i) = \begin{cases} \dfrac{np_0(1-p_0)}{i} & , for\ i < w \\[4mm] \dfrac{np_0(1-p_0)}{w} & , for\ i \geq w \end{cases} \tag{7}$$

and covariance of *MA* is:

$$COV\left(MA_{k_1}, MA_{k_2}\right)$$

$$= \begin{cases} \dfrac{np_0(1-p_0)}{k_2} & for\ k_1, k_2 < w \\[4mm] \dfrac{(k_1 - k_2 + w)}{k_1 w} np_0(1-p_0) & for\ k_1 < w, k_2 \geq w, k_2 - k_1 < w \\[4mm] \dfrac{(k_1 - k_2 + w)}{w^2} np_0(1-p_0) & for\ k_1, k_2 \geq w, k_2 - k_1 < w \\[4mm] 0 & for\ k_2 - k_1 \geq w \end{cases} \tag{8}$$

The center line (*CL*), upper control limit (*UCL*), and lower control limit (*LCL*) of the NPEWMA-MA sign chart are as follows:

$$\left.\begin{aligned} UCL_i &= np_0 + L\sqrt{Var(Z_{S_i})} \\ CL &= np_0 \\ LCL_i &= np_0 - L\sqrt{Var(Z_{S_i})} \end{aligned}\right\} \tag{9}$$

where $L > 0$ is the control limit coefficient selected to achieve a specified IC ARL ($ARL_0$). The mixed EWMA-MA sign chart is computed by plotting the $Z_{S_i}$ with respect to their respective control limits. If $Z_{S_i}$ falls outside the control limit, the process is deemed to be OOC and a quality practitioner should ascertain the assignable cause(s). Conversely, if $LCL_i < Z_{S_i} < UCL_i$, the process is considered IC.

Recently, Haq and Woodall [49] criticized the modified EWMA (mEWMA) control chart proposed by Khan et al. [50], as well as the EWMA-MA control chart and other variants of the EWMA control chart. They argue that these control charts give more importance to past values compared to current ones. They emphasized a specific range of smoothing constants where the performance of the mEWMA chart is lower than that of the traditional EWMA charting scheme. Khan and Aslam [51] updated the results of the mEWMA control chart using Monte Carlo Simulation and showed that the mEWMA control chart is more efficient in terms of ARL than the conventional EWMA control chart. In a similar framework, Alevizakos et al. [52] compared the performance of different variants of the EWMA control chart under same IC RL characteristics. They showed that, for both the zero-state and steady-state cases, these control charts have better OOC RL properties particularly for small to moderate shifts in the process mean. However, the Homogeneously Weighted Moving Average (HWMA) control chart stands out with its deteriorating steady-state OOC RL performance. While the mixed control charts assign more weights to past observations, it is observed that the EWMA-MA charting statistic places more emphasis on the most recent w samples' information, with the weights assigned to earlier observations declining exponentially over time. It is due to the fact that, in the EWMA-MA statistic, the $MA_i$ statistic used in the EWMA charting structure utilizes information from the recent w samples only and ignores the previous ones. Whereas, the

double and triple moving average control charts incorporate the EWMA statistics into other EWMA charting statistic which as a result utilized information from current to first observations repeatedly. Therefore, the EWMA-MA statistic operates similarly to the standard EWMA for observations that are older than w, meaning that their weight drops exponentially. For example, when i = 10, w = 3, and η = 0.3, the total weight assigned to current w ( = 3) sign statistics (S) are 0.489 and the weight assigned to the older samples' information based sign statistics are $w_{S_7} = 0.1533$, $w_{S_6} = 0.10731$, $w_{S_5} = 0.075117$, and so on, where $w_{S_i}$ is the weight assigned to the ith sign statistic ($S_i$). These weights decline exponentially, i.e., each older observation's weight is obtained by multiplying the current one with 1−λ ( = 0.70). We can obtain the weighting strategy of the EWMA-MA charting statistic for any values of i, w, and η by substituting Eq (2) in Eq (4).

## 2.2. The EWMA-MA control chart under arcsine transformation

Mosteller and Youtz [53] recommended the use of arcsine transformation for binomial and Poisson distributions because these distributions are asymmetric for small sample sizes. In line with Yang et al. [26], we apply this transformation to the sign statistic $S_i$ which changes it into a normal random variable $T_i = \sin^{-1}\sqrt{\frac{S_i}{n}}$ with mean $= sin^{-1}\sqrt{p_0}$ and variance $= \frac{1}{4n}$. Now the arcsine NPEWMA-MA chart is constructed as:

$$MA_i = \begin{cases} \dfrac{\sum_{l=1}^{i} T_l}{i} & for\ i < w \\ \dfrac{\sum_{l=i-w+1}^{i} T_l}{w} & for\ i \geq w \end{cases} \tag{10}$$

The plotting statistic of the arcsine NPEWMA-MA control chart is:

$$Z_{T_i} = \eta MA_i + (1 - \eta)Z_{T_{i-1}}, \quad i = 1, 2, 3, \ldots \tag{11}$$

Here, we use the mean of $T$ as an initial value of $Z_T$, i.e., $Z_{T_0} = sin^{-1}\sqrt{p_0} = sin^{-1}\sqrt{0.50}$. The expected value of $Z_{T_i}$ is as follows:

$$E(Z_{T_i}) = \eta \sum_{j=0}^{i-1} (1 - \eta)^j E(MA_{i-j}) + (1 - \eta)^i E(Z_{T_0})$$

$$E(Z_{T_i}) = sin^{-1}\sqrt{p_0} \tag{12}$$

The variance of the $Z_{T_i}$ is:

$$Var(Z_{T_i}) = \eta^2 \sum_{j=0}^{i-1} (1 - \eta)^{2j} var(MA_{i-j})$$
$$+ 2\eta^2 \sum_{k_1=1}^{i-1} \sum_{k_2=k_1+1}^{i} (1 - \eta)^{i-k_1}(1 - \eta)^{i-k_2} Cov(MA_{k_1}, MA_{k_2}) \tag{13}$$

where the variance of MA is defined as:

$$Var(MA_i) = \begin{cases} \dfrac{1}{4ni} & for\ i < w \\ \dfrac{1}{4nw} & for\ i \geq w \end{cases} \tag{14}$$

and covariance of *MA* is:

$$COV\left(MA_{k_1}, MA_{k_2}\right) = \begin{cases} \dfrac{1}{4nk_2} & for\ k_1, k_2 < w \\[2mm] \dfrac{(k_1 - k_2 + w)}{4nk_1 w} & for\ k_1 < w, k_2 \geq w, k_2 - k_1 < w \\[2mm] \dfrac{(k_1 - k_2 + w)}{4nw^2} & for\ k_1, k_2 \geq w, k_2 - k_1 < w \\[2mm] 0 & for\ k_2 - k_1 \geq w \end{cases} \tag{15}$$

The control limits of the arcsine NPEWMA-MA sign chart are respectively as follows:

$$\left. \begin{aligned} UCL_i &= sin^{-1}\sqrt{p_0} + L\sqrt{Var(Z_{T_i})} \\ CL &= sin^{-1}\sqrt{p_0} \\ LCL_i &= sin^{-1}\sqrt{p_0} - L\sqrt{Var(Z_{T_i})} \end{aligned} \right\} \tag{16}$$

The control chart is computed by plotting $Z_{T_i}$ corresponding to their respective control limits given in Eq (16). The process is avowed to be IC if $Z_{T_i}$ falls inside the control limits, and OOC otherwise. This transformation is useful when the desired $ARL_0$ is not achieved due to the discrete nature of the sign statistic. In our case, the obtained $ARL_0$ of the EWMA-MA sign control chart remains within 1% of the desired $ARL_0$, which is quite reasonable. Therefore, both structures can be used interchangeably as the OOC RL performance remains almost the same.

## 3. Performance evaluation

The average run length (ARL), which is the average number of sample points plotted before the first OOC signal occurs, is commonly used to evaluate the performance of a chart. The IC and OOC ARLs are denoted by $ARL_0$ and $ARL_1$, respectively. Since the distribution of *ARL* is skewed, many researchers have criticized the use of *ARL* as a performance measure and suggested using percentiles of RL characteristics to evaluate the performance such as median run length (*MRL*) which is the middle value of the RLs, and standard deviation of run length (*SDRL*) which measures the variability of RLs [54–56]. For a specific $ARL_0$, the control chart with minimum values of $ARL_1$, $MRL_1$, and $SDRL_1$ is more effective in identifying process shifts quickly. We also use average extra quadratic loss (*AEQL*) and relative mean index (*RMI*) for the overall performance evaluation of the suggested chart.

The *AEQL* is based on the loss function and measures the performance of charts over a series of shifts considered in the process. It is defined as:

$$AEQL = \frac{1}{\delta_{max} - \delta_{min}} \sum_{\delta=0}^{\delta_{max}} \delta^2 ARL(\delta) \tag{17}$$

where δ is the amount of shift introduced in the process, ARL(δ) represents the ARL value for a certain shift δ, and $\delta_{min}$ and $\delta_{max}$ are the minimum and maximum shifts considered in the process, respectively. A smaller value of *AEQL* shows the superior performance of the control chart. The *RMI* is recommended by Han and Tsung [57], which depends on the relative differences of the $ARL_1$ and is defined as:

$$RMI = \frac{1}{N} \sum_{i=1}^{N} \left\{ \frac{ARL((\delta_i)) - ARL^*((\delta_i))}{ARL^*((\delta_i))} \right\} \tag{18}$$

where $N$ is the total number of shifts taken into account in the process, $ARL(\delta_i)$ is the chart's $ARL$ value corresponding to a certain shift $\delta_i$, and $ARL^*(\delta_i)$ is the value of a control chart with the lowest $ARL_1$ among all the competing control charts for the specified shift. The control chart with a smaller value of $RMI$ is considered to be more efficient relative to the other control charts.

There are different techniques available in the literature for computing RL profiles of the control charts. The integral equation, Markov chain approach, and Monte Carlo simulation techniques are a few of these. Here, we determine the RL profile of the suggested NPEW-MA-MA chart using Monte Carlo simulation. Compared to other estimation techniques, this method is preferable because it is accurate and versatile enough to handle different scenarios [58].

The simulation study is based on 10,000 replicates using R software. The control charting parameters ($w$, $\eta$, $L$) of the NPEWMA-MA sign chart are selected to achieve a desired $ARL_0$. For a fixed $ARL_0$, the following algorithm is used to calculate the RL profile of the NPEW-MA-MA sign control chart:

Step 1: Generate 10,000 random numbers using the binomial distribution with parameters $n$ and $p$.

Step 2: Choose an arbitrary value of $L$ for fixed values of other design parameters $\eta$ and $w$ to obtain the desired $ARL_0$.

Step 3: Utilize Eq (2) to compute $MA_i$ and subsequently determine the monitoring statistic $Z_{S_i}$.

Step 4: Calculate the control limits, then compare them with charting statistic $Z_{S_i}$.

Step 5: Count the number of samples that fall within the control limits before the EWMA-MA sign control chart triggers the first OOC signal. This count is equal to the single value of RL.

Step 6: Using the same setting, repeat Steps 1 to 5 for 10,000 times to compute the RL characteristics such as $ARL$, $SDRL$, and $MRL$.

$$ARL = \frac{\sum_{i=1}^{N} RL_i}{N} \tag{19}$$

$$SDRL = \sqrt{\frac{\sum_{i=1}^{N} RL_i^2}{N} - (ARL)^2} \tag{20}$$

$$MRL = median(RL) \tag{21}$$

Step7: If the desired $ARL_0$ is attained, then move forward. If not, change the value of $L$ and repeat Steps 1–6 until the desired $ARL_0$ is obtained.

Using the above simulation algorithm, $ARL_0$ is computed by setting $p = p_0 = 0.5$ in Step 1. For $ARL_1$ values, repeat Steps 1 to 6 by setting $p = p_1 \neq 0.5$. In our simulation study, we select $\eta$ = 0.05, 0.10, 0.25, $w$ = 2,3,4,5,8,10, and $n$ = 8(1)20. The above algorithm can also be used for the arcsine EWMA-MA control chart after using the transformation discussed in Section 2.2.

Table 1 presents the values of the control limit coefficient ($L$) of the proposed NPEW-MA-MA sign control chart for different sample sizes and various combinations of design parameters (w, L, η) under nominal ARL₀≅370. For a fixed value of η and n, it can be noticed that the value of the limit coefficient (L) declines as the value of the span (w) increases. It is also observed that the value of the value of L increases with η for a fixed value of n and w.

To study the OOC RL performance of the NPEWMA-MA chart for various shifts in the process proportion, we consider different values of sample size and design parameters $\eta$ and

**Table 1. The $L$ values of the EWMA-MA sign chart for various combinations of $(n, w, \eta)$ at $ARL_0 \cong 370$.**

| $\eta$ | $w$ | $n$ | | | | | | | | | | | | |
|---|---|---|---|---|---|---|---|---|---|---|---|---|---|---|
| | | **8** | **9** | **10** | **11** | **12** | **13** | **14** | **15** | **16** | **17** | **18** | **19** | **20** |
| 0.05 | 2 | 2.418 | 2.414 | 2.422 | 2.419 | 2.420 | 2.421 | 2.422 | 2.423 | 2.421 | 2.422 | 2.420 | 2.424 | 2.425 |
| | 3 | 2.370 | 2.371 | 2.372 | 2.374 | 2.369 | 2.371 | 2.372 | 2.374 | 2.373 | 2.374 | 2.376 | 2.372 | 2.377 |
| | 4 | 2.335 | 2.333 | 2.336 | 2.337 | 2.338 | 2.334 | 2.335 | 2.337 | 2.336 | 2.339 | 2.339 | 2.340 | 2.341 |
| | 5 | 2.308 | 2.306 | 2.305 | 2.307 | 2.303 | 2.306 | 2.310 | 2.309 | 2.312 | 2.314 | 2.308 | 2.304 | 2.305 |
| | 8 | 2.232 | 2.238 | 2.240 | 2.240 | 2.241 | 2.239 | 2.236 | 2.235 | 2.236 | 2.238 | 2.237 | 2.236 | 2.234 |
| | 10 | 2.199 | 2.201 | 2.203 | 2.200 | 2.197 | 2.199 | 2.198 | 2.203 | 2.205 | 2.203 | 2.204 | 2.203 | 2.200 |
| 0.10 | 2 | 2.613 | 2.610 | 2.610 | 2.615 | 2.614 | 2.616 | 2.617 | 2.619 | 2.620 | 2.616 | 2.618 | 2.620 | 2.621 |
| | 3 | 2.556 | 2.555 | 2.560 | 2.557 | 2.559 | 2.560 | 2.561 | 2.562 | 2.563 | 2.564 | 2.565 | 2.562 | 2.563 |
| | 4 | 2.513 | 2.511 | 2.516 | 2.517 | 2.519 | 2.520 | 2.521 | 2.522 | 2.523 | 2.524 | 2.525 | 2.521 | 2.523 |
| | 5 | 2.483 | 2.480 | 2.480 | 2.484 | 2.485 | 2.487 | 2.486 | 2.487 | 2.485 | 2.486 | 2.487 | 2.488 | 2.489 |
| | 8 | 2.405 | 2.403 | 2.408 | 2.409 | 2.410 | 2.411 | 2.413 | 2.410 | 2.405 | 2.407 | 2.408 | 2.409 | 2.410 |
| | 10 | 2.367 | 2.365 | 2.365 | 2.366 | 2.364 | 2.365 | 2.366 | 2.367 | 2.368 | 2.369 | 2.370 | 2.367 | 2.369 |
| 0.25 | 2 | 2.785 | 2.783 | 2.787 | 2.790 | 2.794 | 2.796 | 2.799 | 2.801 | 2.804 | 2.806 | 2.807 | 2.808 | 2.809 |
| | 3 | 2.727 | 2.737 | 2.741 | 2.742 | 2.743 | 2.745 | 2.746 | 2.748 | 2.749 | 2.750 | 2.749 | 2.751 | 2.750 |
| | 4 | 2.691 | 2.693 | 2.695 | 2.696 | 2.697 | 2.698 | 2.700 | 2.702 | 2.704 | 2.706 | 2.707 | 2.706 | 2.707 |
| | 5 | 2.659 | 2.660 | 2.663 | 2.662 | 2.664 | 2.665 | 2.666 | 2.667 | 2.668 | 2.668 | 2.669 | 2.670 | 2.671 |
| | 8 | 2.580 | 2.582 | 2.579 | 2.581 | 2.583 | 2.586 | 2.587 | 2.588 | 2.590 | 2.591 | 2.592 | 2.590 | 2.587 |
| | 10 | 2.552 | 2.545 | 2.546 | 2.547 | 2.549 | 2.550 | 2.554 | 2.551 | 2.550 | 2.549 | 2.550 | 2.553 | 2.559 |

$w$. The limit coefficient value ($L$) is chosen from Table 1 to get the desired $ARL_0 \cong 370$. The results shown in Tables 2–5 are summarized as follows:

i. The OOC RL profile ($ARL_1$, $SDRL_1$, $MRL_1$) of the NPEWMA-MA sign chart decreases as the sample size $n$ increases.

ii. For fixed values of $\eta$, $w$, and $n$, it is observed that the values of $ARL_1$, $SDRL_1$, and $MRL_1$ decrease rapidly as the size of the shift in process proportion ($\Delta$), i.e., $\Delta = |p_0 - p_1|$, increases.

iii. For minor to moderate shifts in the process proportion, both the $ARL_1$ and $MRL_1$ decline as the value of $w$ increases.

iv. Moreover, for smaller values of the smoothing parameter $\eta$, the NPEWMA-MA sign chart has a superior shift recognition ability.

v. The RL distribution of the NPEWMA-MA chart is positively skewed as $ARL_0 > MRL_0$.

Generally, a large value of $w$ and a smaller value of smoothing parameter $\eta$ is recommended if quick detection of small shifts is desirable.

To study the performance and robustness of the proposed chart, various symmetrical and skewed distributions are considered such as the standard normal distribution $N(0,1)$; the Weibull distribution, $Weibull(2,1)$ and $Weibull(3.5,1)$; the Logistic distribution, $LG\left(0, \frac{\sqrt{3}}{\pi}\right)$; the Student's t distribution, $t(5)$ and $t(10)$; the Laplace distribution, $Laplace\left(0, \frac{1}{\sqrt{2}}\right)$; the Gamma distribution, $gamma(2,1)$ and $gamma(5,1)$; and the contaminated normal ($CN$) distribution. The CN distribution is the combination of two normal distributions having a common mean $\mu$ but different variances which are formulated as $(1 - \beta)N(\mu, \sigma_1^2) + \beta N(\mu, \sigma_2^2)$. For CN distribution, we assume $\sigma_1 = 2\sigma_2$ and level of contamination ($\beta$) is equal to 0.10. For $ARL_0 \cong 370$, $n = 10$, and different combinations of design parameters ($\eta$, $w$), the RL profile of the above-listed distributions are presented in Tables 6, 7 and summarized as follows:

**Table 2. The RL profile of the EWMA-MA sign chart at $ARL_0 \cong 370$ with $\eta = 0.05$ and $w = 5$.**

| $p_1$ | n | | | | | | | | | | | | | | |
|---|---|---|---|---|---|---|---|---|---|---|---|---|---|---|---|
| | 8 | | | 10 | | | 12 | | | 15 | | | 20 | | |
| | ARL | MRL | SDRL | ARL | MRL | SDRL | ARL | MRL | SDRL | ARL | MRL | SDRL | ARL | MRL | SDRL |
| 0.05 | 1.4 | 1 | 0.6 | 1.1 | 1 | 0.3 | 1.0 | 1 | 0.2 | 1.0 | 1 | 0.1 | 1.0 | 1 | 0.1 |
| 0.10 | 1.8 | 2 | 0.9 | 1.3 | 1 | 0.6 | 1.1 | 1 | 0.5 | 1.1 | 1 | 0.3 | 1.0 | 1 | 0.2 |
| 0.15 | 2.4 | 2 | 1.4 | 1.8 | 1 | 1.1 | 1.5 | 1 | 0.9 | 1.3 | 1 | 0.7 | 1.2 | 1 | 0.4 |
| 0.20 | 3.3 | 3 | 1.9 | 2.5 | 2 | 1.7 | 2.0 | 1 | 1.5 | 1.7 | 1 | 1.1 | 1.5 | 1 | 0.8 |
| 0.25 | 4.5 | 4 | 2.6 | 3.7 | 3 | 2.4 | 3.0 | 3 | 2.2 | 2.5 | 2 | 1.8 | 2.1 | 2 | 1.2 |
| 0.30 | 6.5 | 6 | 3.8 | 5.4 | 5 | 3.3 | 4.5 | 4 | 3.0 | 3.8 | 3 | 2.6 | 3.1 | 3 | 2.0 |
| 0.35 | 10.0 | 9 | 6.0 | 8.5 | 8 | 5.1 | 7.2 | 7 | 4.6 | 6.1 | 6 | 3.8 | 5.0 | 5 | 3.1 |
| 0.40 | 18.8 | 17 | 12.5 | 15.6 | 14 | 10.2 | 13.4 | 12 | 8.9 | 11.4 | 10 | 7.4 | 9.3 | 9 | 5.6 |
| 0.45 | 52.9 | 46 | 43.1 | 46.8 | 37 | 37.8 | 39.0 | 31 | 32.0 | 34.6 | 25 | 22.4 | 26.9 | 22 | 19.9 |
| 0.50 | 371.5 | 246 | 356.9 | 371.7 | 258 | 365.5 | 370.1 | 262 | 357.9 | 371 | 259 | 363.7 | 370.6 | 261 | 366 |
| 0.55 | 53.8 | 46 | 43.8 | 46.2 | 37 | 38.0 | 39.5 | 31 | 32.3 | 35.3 | 25 | 22.9 | 26.6 | 22 | 19.4 |
| 0.60 | 18.8 | 17 | 12.4 | 15.8 | 14 | 10.5 | 13.4 | 12 | 8.8 | 11.4 | 10 | 7.2 | 9.2 | 9 | 5.6 |
| 0.65 | 10.0 | 9 | 5.9 | 8.5 | 8 | 5.2 | 7.2 | 7 | 4.5 | 6.1 | 6 | 3.8 | 5.1 | 5 | 3.1 |
| 0.70 | 6.5 | 6 | 3.7 | 5.4 | 5 | 3.3 | 4.6 | 4 | 3 | 3.8 | 3 | 2.6 | 3.1 | 3 | 2.0 |
| 0.75 | 4.5 | 4 | 2.6 | 3.7 | 3 | 2.4 | 3.0 | 3 | 2.1 | 2.5 | 2 | 1.8 | 2.1 | 2 | 1.3 |
| 0.80 | 3.3 | 3 | 1.9 | 2.5 | 2 | 1.7 | 2.0 | 1 | 1.5 | 1.7 | 1 | 1.1 | 1.5 | 1 | 0.8 |
| 0.85 | 2.4 | 2 | 1.4 | 1.8 | 1 | 1.1 | 1.5 | 1 | 0.9 | 1.3 | 1 | 0.6 | 1.2 | 1 | 0.4 |
| 0.90 | 1.8 | 2 | 0.9 | 1.3 | 1 | 0.6 | 1.2 | 1 | 0.5 | 1.1 | 1 | 0.3 | 1.0 | 1 | 0.2 |
| 0.95 | 1.4 | 1 | 0.6 | 1.1 | 1 | 0.3 | 1.0 | 1 | 0.2 | 1.0 | 1 | 0.1 | 1.0 | 1 | 0 |

**Table 3. The RL profile of the EWMA-MA sign chart at $ARL_0 \cong 370$ with $\eta = 0.05$ and $w = 10$.**

| $p_1$ | n | | | | | | | | | | | | | | |
|---|---|---|---|---|---|---|---|---|---|---|---|---|---|---|---|
| | 8 | | | 10 | | | 12 | | | 15 | | | 20 | | |
| | ARL | MRL | SDRL | ARL | MRL | SDRL | ARL | MRL | SDRL | ARL | MRL | SDRL | ARL | MRL | SDRL |
| 0.05 | 1.4 | 1 | 0.5 | 1.1 | 1 | 0.3 | 1.0 | 1 | 0.1 | 1.0 | 1 | 0.1 | 1.0 | 1 | 0.0 |
| 0.10 | 1.8 | 2 | 0.8 | 1.3 | 1 | 0.6 | 1.1 | 1 | 0.4 | 1.1 | 1 | 0.3 | 1.0 | 1 | 0.1 |
| 0.15 | 2.3 | 2 | 1.3 | 1.7 | 1 | 1.0 | 1.4 | 1 | 0.8 | 1.3 | 1 | 0.6 | 1.1 | 1 | 0.3 |
| 0.20 | 3.1 | 2 | 2.1 | 2.4 | 2 | 1.7 | 1.9 | 1 | 1.4 | 1.6 | 1 | 1.1 | 1.3 | 1 | 0.7 |
| 0.25 | 4.5 | 4 | 3.1 | 3.6 | 3 | 2.6 | 2.9 | 2 | 2.3 | 2.4 | 2 | 1.8 | 1.7 | 1 | 1.2 |
| 0.30 | 6.8 | 6 | 4.5 | 5.5 | 5 | 3.9 | 4.5 | 4 | 3.5 | 3.7 | 3 | 2.8 | 2.8 | 2 | 2.1 |
| 0.35 | 10.8 | 11 | 6.7 | 9.1 | 9 | 5.9 | 7.7 | 7 | 5.4 | 6.5 | 6 | 4.6 | 5.0 | 4 | 3.7 |
| 0.40 | 19.6 | 18 | 12.9 | 16.4 | 15 | 10.6 | 14.2 | 14 | 9.3 | 12.1 | 12 | 7.8 | 9.8 | 10 | 6.5 |
| 0.45 | 50.1 | 39 | 41.2 | 42.5 | 32 | 38 | 36.5 | 30 | 28.7 | 30.9 | 25 | 23.5 | 22.3 | 20 | 16.3 |
| 0.50 | 371.4 | 260 | 374.0 | 370.9 | 255 | 383.5 | 372.6 | 255 | 386.0 | 369.5 | 254 | 373.1 | 370.4 | 252 | 380.1 |
| 0.55 | 50.5 | 38 | 42.8 | 43.0 | 33 | 38.2 | 37.3 | 30 | 29.3 | 31.2 | 25 | 23.9 | 22.9 | 21 | 16.9 |
| 0.60 | 19.7 | 18 | 12.9 | 16.5 | 15 | 10.6 | 14.4 | 14 | 9.3 | 12.3 | 12 | 7.7 | 9.8 | 9 | 6.5 |
| 0.65 | 10.8 | 11 | 6.8 | 9.1 | 8 | 5.9 | 7.7 | 7 | 5.3 | 6.4 | 6 | 4.6 | 4.9 | 4 | 3.7 |
| 0.70 | 6.8 | 6 | 4.5 | 5.5 | 5 | 3.9 | 4.6 | 4 | 3.5 | 3.8 | 3 | 2.8 | 2.8 | 2 | 2.2 |
| 0.75 | 4.5 | 4 | 3.1 | 3.6 | 3 | 2.6 | 2.9 | 2 | 2.3 | 2.4 | 2 | 1.7 | 1.8 | 1 | 1.2 |
| 0.80 | 3.1 | 2 | 2.1 | 2.4 | 2 | 1.7 | 1.9 | 1 | 1.4 | 1.6 | 1 | 1.1 | 1.3 | 1 | 0.7 |
| 0.85 | 2.3 | 2 | 1.3 | 1.8 | 1 | 1.1 | 1.4 | 1 | 0.8 | 1.2 | 1 | 0.6 | 1.1 | 1 | 0.3 |
| 0.90 | 1.8 | 2 | 0.8 | 1.3 | 1 | 0.6 | 1.1 | 1 | 0.4 | 1.1 | 1 | 0.3 | 1.0 | 1 | 0.1 |
| 0.95 | 1.4 | 1 | 0.6 | 1.1 | 1 | 0.3 | 1.0 | 1 | 0.2 | 1.0 | 1 | 0.1 | 1.0 | 1 | 0.0 |

**Table 4. The RL profile of the EWMA-MA sign chart at $ARL_0 \cong 370$ with $\eta = 0.10$ and $w = 5$.**

| $p_1$ | $n$ | | | | | | | | | | | | | | |
|---|---|---|---|---|---|---|---|---|---|---|---|---|---|---|---|
| | 8 | | | 10 | | | 12 | | | 15 | | | 20 | | |
| | ARL | MRL | SDRL | ARL | MRL | SDRL | ARL | MRL | SDRL | ARL | MRL | SDRL | ARL | MRL | SDRL |
| 0.05 | 1.4 | 1 | 0.7 | 1.1 | 1 | 0.4 | 1.1 | 1 | 0.3 | 1.0 | 1 | 0.2 | 1.0 | 1 | 0.0 |
| 0.10 | 2.0 | 2 | 1.1 | 1.4 | 1 | 0.8 | 1.4 | 1 | 0.6 | 1.2 | 1 | 0.5 | 1.0 | 1 | 0.2 |
| 0.15 | 2.6 | 2 | 1.5 | 2.0 | 1 | 1.3 | 1.8 | 2 | 1.0 | 1.5 | 1 | 0.8 | 1.2 | 1 | 0.5 |
| 0.20 | 3.6 | 3 | 2.0 | 2.8 | 3 | 1.8 | 2.5 | 2 | 1.4 | 2.1 | 2 | 1.2 | 1.5 | 1 | 0.8 |
| 0.25 | 4.9 | 5 | 2.6 | 3.9 | 4 | 2.4 | 3.5 | 3 | 2.0 | 2.9 | 3 | 1.7 | 2.2 | 2 | 1.3 |
| 0.30 | 6.9 | 7 | 3.7 | 5.7 | 6 | 3.3 | 5.1 | 5 | 2.8 | 4.3 | 4 | 2.4 | 3.3 | 3 | 2.1 |
| 0.35 | 10.8 | 10 | 6.3 | 8.9 | 8 | 5.3 | 7.9 | 8 | 4.4 | 6.7 | 7 | 3.7 | 5.3 | 5 | 3.1 |
| 0.40 | 20.6 | 17 | 14.9 | 16.8 | 14 | 11.7 | 14.7 | 13 | 9.6 | 12.3 | 11 | 7.7 | 9.9 | 9 | 5.9 |
| 0.45 | 64.8 | 45 | 56.4 | 56.8 | 42 | 49.4 | 49.4 | 37 | 42.9 | 40.1 | 30 | 34.0 | 30.9 | 24 | 24.4 |
| 0.50 | 372.5 | 259 | 366.8 | 372.3 | 258 | 369.9 | 370.4 | 259 | 374.2 | 371.2 | 255 | 375.6 | 369.7 | 259 | 369.7 |
| 0.55 | 65.6 | 47 | 59.8 | 56.7 | 42 | 50.8 | 48.7 | 36 | 42.0 | 40.8 | 31 | 34.4 | 31.2 | 24 | 24.7 |
| 0.60 | 20.9 | 17 | 15.1 | 17.1 | 14 | 12.0 | 14.9 | 13 | 9.8 | 12.4 | 11 | 7.7 | 9.9 | 9 | 5.8 |
| 0.65 | 10.7 | 10 | 6.3 | 8.9 | 8 | 5.2 | 7.9 | 8 | 4.4 | 6.7 | 6 | 3.7 | 5.4 | 5 | 3.1 |
| 0.70 | 6.9 | 7 | 3.8 | 5.7 | 6 | 3.3 | 5.1 | 5 | 2.8 | 4.2 | 4 | 2.4 | 3.3 | 3 | 2.0 |
| 0.75 | 4.8 | 5 | 2.6 | 3.9 | 4 | 2.4 | 3.5 | 3 | 2.0 | 2.9 | 3 | 1.7 | 2.2 | 2 | 1.3 |
| 0.80 | 3.6 | 3 | 2.0 | 2.8 | 3 | 1.8 | 2.5 | 2 | 1.4 | 2.1 | 2 | 1.2 | 1.5 | 1 | 0.8 |
| 0.85 | 2.6 | 2 | 1.5 | 2.0 | 1 | 1.3 | 1.8 | 2 | 1.0 | 1.5 | 1 | 0.8 | 1.2 | 1 | 0.5 |
| 0.90 | 2.0 | 2 | 1.1 | 1.4 | 1 | 0.8 | 1.4 | 1 | 0.6 | 1.2 | 1 | 0.5 | 1.0 | 1 | 0.2 |
| 0.95 | 1.4 | 1 | 0.7 | 1.1 | 1 | 0.4 | 1.1 | 1 | 0.3 | 1.0 | 1 | 0.2 | 1.0 | 1 | 0.0 |

**Table 5. The RL profile of the EWMA-MA sign chart at $ARL_0 \cong 370$ with $\eta = 0.10$ and $w = 10$.**

| $p_1$ | $n$ | | | | | | | | | | | | | | | | | |
|---|---|---|---|---|---|---|---|---|---|---|---|---|---|---|---|---|---|---|
| | 8 | | | 10 | | | 12 | | | 15 | | | 18 | | | 20 | | |
| | ARL | MRL | SDRL | ARL | MRL | SDRL | ARL | MRL | SDRL | ARL | MRL | SDRL | ARL | MRL | SDRL | ARL | MRL | SDRL |
| 0.05 | 1.4 | 1 | 0.6 | 1.1 | 1 | 0.3 | 1.1 | 1 | 0.3 | 1.0 | 1 | 0.2 | 1.0 | 1 | 0.1 | 1.0 | 1 | 0.0 |
| 0.10 | 1.9 | 2 | 1.0 | 1.4 | 1 | 0.7 | 1.4 | 1 | 0.6 | 1.2 | 1 | 0.4 | 1.1 | 1 | 0.3 | 1.0 | 1 | 0.2 |
| 0.15 | 2.4 | 2 | 1.4 | 1.8 | 1 | 1.1 | 1.8 | 2 | 0.9 | 1.5 | 1 | 0.7 | 1.3 | 1 | 0.6 | 1.2 | 1 | 0.4 |
| 0.20 | 3.4 | 3 | 2.1 | 2.6 | 2 | 1.8 | 2.3 | 2 | 1.4 | 2.0 | 2 | 1.1 | 1.7 | 1 | 0.9 | 1.5 | 1 | 0.8 |
| 0.25 | 4.8 | 4 | 3.0 | 3.8 | 3 | 2.6 | 3.3 | 3 | 2.1 | 2.8 | 2 | 1.7 | 2.3 | 2 | 1.4 | 2.1 | 2 | 1.3 |
| 0.30 | 7.1 | 7 | 4.3 | 5.8 | 5 | 3.9 | 5.1 | 4 | 3.3 | 4.1 | 3 | 2.7 | 3.5 | 3 | 2.3 | 3.2 | 3 | 2.1 |
| 0.35 | 11.2 | 11 | 6.6 | 9.3 | 9 | 5.7 | 8.3 | 8 | 5.0 | 7.0 | 6 | 4.3 | 6.0 | 5 | 3.8 | 5.4 | 5 | 3.5 |
| 0.40 | 20.9 | 18 | 14.2 | 17.6 | 16 | 11.6 | 15.1 | 14 | 9.5 | 12.9 | 12 | 8.0 | 11.3 | 11 | 6.8 | 10.4 | 10 | 6.3 |
| 0.45 | 60.2 | 44 | 58.1 | 51.4 | 38 | 47.6 | 44.5 | 32 | 40.9 | 35.7 | 27 | 31.7 | 30.8 | 24 | 25.7 | 28.3 | 22 | 24.2 |
| 0.50 | 372.7 | 255 | 388.3 | 370.2 | 250 | 387.6 | 371.5 | 250 | 389.3 | 371.3 | 248 | 397.9 | 371.5 | 248 | 393.7 | 370.7 | 245 | 391.5 |
| 0.55 | 60.7 | 44 | 57.8 | 51.9 | 38 | 48.5 | 44.4 | 33 | 40.9 | 36 | 27 | 32.4 | 30.7 | 23 | 26.6 | 27.9 | 21 | 24.1 |
| 0.60 | 21.3 | 18 | 14.4 | 17.5 | 16 | 11.4 | 15.0 | 14 | 9.4 | 12.9 | 12 | 7.8 | 11.2 | 11 | 6.7 | 10.4 | 10 | 6.1 |
| 0.65 | 11.2 | 11 | 6.6 | 9.3 | 9 | 5.7 | 8.2 | 8 | 4.9 | 6.9 | 6 | 4.3 | 6.0 | 5 | 3.8 | 5.5 | 5 | 3.6 |
| 0.70 | 7.2 | 7 | 4.3 | 5.9 | 5 | 3.9 | 5.1 | 4 | 3.3 | 4.2 | 4 | 2.7 | 3.5 | 3 | 2.3 | 3.2 | 3 | 2.1 |
| 0.75 | 4.8 | 4 | 3.0 | 3.8 | 3 | 2.7 | 3.3 | 3 | 2.1 | 2.8 | 2 | 1.7 | 2.4 | 2 | 1.4 | 2.1 | 2 | 1.2 |
| 0.80 | 3.4 | 3 | 2.1 | 2.6 | 2 | 1.8 | 2.3 | 2 | 1.4 | 2.0 | 2 | 1.1 | 1.7 | 1 | 0.9 | 1.5 | 1 | 0.8 |
| 0.85 | 2.5 | 2 | 1.4 | 1.8 | 1 | 1.2 | 1.8 | 2 | 0.9 | 1.5 | 1 | 0.7 | 1.3 | 1 | 0.5 | 1.2 | 1 | 0.4 |
| 0.90 | 1.9 | 2 | 0.9 | 1.4 | 1 | 0.7 | 1.4 | 1 | 0.6 | 1.2 | 1 | 0.4 | 1.1 | 1 | 0.3 | 1.0 | 1 | 0.2 |
| 0.95 | 1.4 | 1 | 0.6 | 1.1 | 1 | 0.4 | 1.1 | 1 | 0.3 | 1.0 | 1 | 0.2 | 1.0 | 1 | 0.1 | 1.0 | 1 | 0.1 |

**Table 6. The RL characteristics of the EWMA-MA sign control chart under various distributions for $\eta = 0.05$, $w = 5$, $n = 10$, and $L = 2.305$ at $ARL_0 \cong 370$.**

| Distribution | Characteristic | $\delta$ | | | | | | | | | | | AEQL |
|---|---|---|---|---|---|---|---|---|---|---|---|---|---|
| | | 0 | 0.05 | 0.10 | 0.25 | 0.50 | 0.75 | 1.00 | 1.50 | 2.00 | 2.5 | 3.0 | |
| $N(0,1)$ | ARL | 368.7 | 170.2 | 66.3 | 15.7 | 5.8 | 3.1 | 1.9 | 1.2 | 1.0 | 1.0 | 1.0 | **9.7** |
| | SDRL | 357.3 | 160.9 | 55.1 | 10.4 | 3.6 | 2.1 | 1.3 | 0.5 | 0.2 | 0.0 | 0.0 | |
| | MRL | 260 | 122 | 52 | 14 | 6 | 3 | 1 | 1 | 1 | 1 | 1 | |
| $Weibull(2,1)$ | ARL | 369.2 | 192.5 | 81.7 | 21.3 | 8.6 | 5.4 | 3.9 | 2.5 | 1.9 | 1.5 | 1.4 | **15.6** |
| | SDRL | 358.7 | 180.7 | 72.5 | 15.0 | 5.3 | 3.3 | 2.5 | 1.7 | 1.2 | 0.9 | 0.7 | |
| | MRL | 262 | 137 | 59 | 18 | 8 | 5 | 4 | 2 | 1 | 1 | 1 | |
| $Weibull(3.5,1)$ | ARL | 371.7 | 178.6 | 72.4 | 19.1 | 7.6 | 4.6 | 3.2 | 2.0 | 1.5 | 1.3 | 1.2 | **13.2** |
| | SDRL | 358.0 | 171.9 | 65.6 | 13.0 | 4.6 | 2.9 | 2.1 | 1.3 | 0.8 | 0.6 | 0.4 | |
| | MRL | 261 | 125 | 53 | 16 | 7 | 4 | 3 | 1 | 1 | 1 | 1 | |
| $LG\left(0,\frac{\sqrt{3}}{\pi}\right)$ | ARL | 368.5 | 149.7 | 54.2 | 13.4 | 4.9 | 2.6 | 1.7 | 1.1 | 1 | 1 | 1 | **9.3** |
| | SDRL | 361.4 | 137.8 | 43.7 | 8.6 | 3.1 | 1.8 | 1.1 | 0.4 | 0.2 | 0.1 | 0 | |
| | MRL | 259 | 108 | 43 | 12 | 5 | 2 | 1 | 1 | 1 | 1 | 1 | |
| $CN$ | ARL | 370.1 | 152.8 | 67.1 | 13.8 | 5.1 | 2.8 | 1.8 | 1.1 | 1.0 | 1.0 | 1.0 | **9.4** |
| | SDRL | 364.1 | 143.7 | 48.7 | 9.0 | 3.2 | 1.9 | 1.1 | 0.4 | 0.1 | 0.0 | 0.0 | |
| | MRL | 258 | 114 | 43 | 12 | 5 | 2 | 1 | 1 | 1 | 1 | 1 | |
| $t(5)$ | ARL | 369.8 | 133.6 | 49.2 | 11.8 | 4.3 | 2.4 | 1.6 | 1.1 | 1.0 | 1.0 | 1.0 | **9.1** |
| | SDRL | 365.7 | 121.6 | 30.1 | 7.5 | 2.8 | 1.6 | 0.9 | 0.4 | 0.2 | 0.1 | 0.0 | |
| | MRL | 256 | 95 | 39 | 11 | 4 | 2 | 1 | 1 | 1 | 1 | 1 | |
| $t(10)$ | ARL | 371.3 | 155.5 | 58.2 | 14.1 | 5.1 | 2.7 | 1.8 | 1.2 | 1.0 | 1.0 | 1.0 | **95** |
| | SDRL | 364.1 | 144.3 | 49.8 | 9.2 | 3.2 | 1.8 | 1.1 | 0.4 | 0.2 | 0.1 | 0.0 | |
| | MRL | 260 | 112 | 44 | 12 | 5 | 2 | 1 | 1 | 1 | 1 | 1 | |
| $Laplace\left(0,\frac{1}{\sqrt{2}}\right)$ | ARL | 372.7 | 87.2 | 29.8 | 8.6 | 3.6 | 2.1 | 1.5 | 1.1 | 1.0 | 1.0 | 1.0 | **8.8** |
| | SDRL | 362.8 | 74.8 | 22.4 | 5.2 | 2.3 | 1.4 | 0.9 | 0.4 | 0.2 | 0.1 | 0.0 | |
| | MRL | 260 | 65 | 25 | 8 | 3 | 2 | 1 | 1 | 1 | 1 | 1 | |
| $gamma(2,1)$ | ARL | 369.4 | 153.0 | 58.8 | 14.5 | 5.5 | 3.1 | 2.0 | 1.2 | 1.0 | 1.0 | 1.0 | **9.6** |
| | SDRL | 365.1 | 144.5 | 51.2 | 9.4 | 3.4 | 2.0 | 1.3 | 0.5 | 0.2 | 0.1 | 0.0 | |
| | MRL | 261 | 112 | 45 | 13 | 5 | 3 | 2 | 1 | 1 | 1 | 1 | |
| $gamma(5,1)$ | ARL | 370.1 | 165.7 | 64.4 | 15.3 | 5.7 | 3.2 | 2.0 | 1.2 | 1.0 | 1.0 | 1.0 | **9.7** |
| | SDRL | 354.9 | 153.4 | 55.2 | 10.0 | 3.5 | 2.1 | 1.3 | 0.5 | 0.2 | 0.1 | 0.0 | |
| | MRL | 264 | 119 | 48 | 14 | 6 | 3 | 2 | 1 | 1 | 1 | 1 | |

i. The results signify that for all continuous distributions taken into account in this study, the IC RL properties of the proposed NPEWMA-MA sign control chart are the same which is in line with the definition of a nonparametric control chart.

ii. As the magnitude of shift increases ($\delta$), the OOC RL profile declines.

iii. For small to moderate shifts, the OOC RL characteristics tend to increases with $\eta$ under fixed value of $n$ and $w$.

iv. Unlike other distributions, the proposed chart works efficiently when the distribution of the underlying process is Laplace. In addition, the overall performance measurement value of $AEQL$ is minimal for the Laplace distribution.

## 4. Comparison study

The NPEWMA-MA sign control chart is compared with some existing nonparametric control charts such as the MA sign control chart suggested by Pawar et al. [30], the EWMA sign

**Table 7. The RL characteristics of the EWMA-MA sign control chart under various distributions for $\eta = 0.10$, $w = 5$, $n = 10$, and $L = 2.480$ at $ARL_0 \cong 370$.**

| Distribution | Characteristic | $\delta$ | | | | | | | | | | | AEQL |
|---|---|---|---|---|---|---|---|---|---|---|---|---|---|
| | | 0 | 0.05 | 0.10 | 0.25 | 0.50 | 0.75 | 1.00 | 1.50 | 2.00 | 2.5 | 3.0 | |
| $N(0,1)$ | ARL | 372.3 | 202.2 | 82.8 | 17.5 | 6.1 | 3.3 | 2.1 | 1.2 | 1.0 | 1.0 | 1.0 | **9.9** |
| | SDRL | 368.3 | 197.2 | 77.3 | 12.1 | 3.5 | 2.1 | 1.3 | 0.5 | 0.2 | 0.0 | 0.0 | |
| | MRL | 256.5 | 142 | 59 | 15 | 6 | 3 | 1 | 1 | 1 | 1 | 1 | |
| $Weibull(2,1)$ | ARL | 369.6 | 222.2 | 101.8 | 24.3 | 9.2 | 5.7 | 4.2 | 2.7 | 2.0 | 1.6 | 1.4 | **16.5** |
| | SDRL | 365.4 | 218.5 | 93.8 | 18.3 | 5.4 | 3.3 | 2.5 | 1.8 | 1.3 | 1.0 | 0.8 | |
| | MRL | 254 | 157 | 72 | 20 | 9 | 6 | 4 | 2 | 1 | 1 | 1 | |
| $Weibull(3.5,1)$ | ARL | 370.5 | 209.2 | 90.7 | 21.1 | 8.0 | 4.9 | 3.5 | 2.1 | 1.6 | 1.3 | 1.2 | **13.7** |
| | SDRL | 368.6 | 205.2 | 83.7 | 15.6 | 4.6 | 2.9 | 2.2 | 1.4 | 1.0 | 0.7 | 0.5 | |
| | MRL | 253 | 145 | 66 | 17 | 8 | 5 | 3 | 2 | 1 | 1 | 1 | |
| $LG\left(0,\frac{\sqrt{3}}{\pi}\right)$ | ARL | 372.8 | 177.4 | 67.8 | 14.1 | 5.2 | 2.9 | 1.8 | 1.2 | 1.0 | 1.0 | 1.0 | **9.6** |
| | SDRL | 365.6 | 175.9 | 60.3 | 9.3 | 3.0 | 1.9 | 1.2 | 0.5 | 0.2 | 0.1 | 0.0 | |
| | MRL | 262 | 122 | 50 | 12 | 5 | 3 | 1 | 1 | 1 | 1 | 1 | |
| $CN$ | ARL | 373.3 | 182.8 | 71.3 | 14.9 | 5.4 | 3.0 | 1.9 | 1.1 | 1.0 | 1.0 | 1.0 | **9.6** |
| | SDRL | 372.2 | 177.5 | 62.7 | 10.0 | 3.1 | 1.9 | 1.2 | 0.4 | 0.1 | 0.0 | 0.0 | |
| | MRL | 257 | 129 | 52 | 13 | 5 | 3 | 1 | 1 | 1 | 1 | 1 | |
| $t(5)$ | ARL | 368.4 | 160.9 | 59.3 | 12.8 | 4.7 | 2.6 | 1.7 | 1.1 | 1.0 | 1.0 | 1.0 | **9.3** |
| | SDRL | 367.9 | 153.2 | 52.5 | 8.1 | 2.8 | 1.7 | 1.0 | 0.4 | 0.2 | 0.1 | 0.0 | |
| | MRL | 254 | 115 | 44 | 11 | 5 | 2 | 1 | 1 | 1 | 1 | 1 | |
| $t(10)$ | ARL | 372.1 | 183.8 | 72.3 | 15.2 | 5.5 | 3 | 1.9 | 1.2 | 1.0 | 1.0 | 1.0 | **9.7** |
| | SDRL | 374.7 | 179.6 | 65.7 | 10.1 | 3.1 | 1.9 | 1.2 | 0.5 | 0.2 | 0.1 | 0.0 | |
| | MRL | 258 | 131 | 53 | 13 | 6 | 3 | 1 | 1 | 1 | 1 | 1 | |
| $Laplace\left(0,\frac{1}{\sqrt{2}}\right)$ | ARL | 369.5 | 102.9 | 34.8 | 9.0 | 3.8 | 2.3 | 1.6 | 1.2 | 1.0 | 1.0 | 1.0 | **8.9** |
| | SDRL | 370.1 | 98.5 | 28 | 5.3 | 2.4 | 1.5 | 1.0 | 0.5 | 0.2 | 0.1 | 0.0 | |
| | MRL | 254 | 73 | 27 | 8 | 4 | 2 | 1 | 1 | 1 | 1 | 1 | |
| $gamma(2,1)$ | ARL | 372.5 | 183.6 | 72.2 | 15.4 | 5.8 | 3.3 | 2.1 | 1.3 | 1.1 | 1.0 | 1.0 | **10.0** |
| | SDRL | 370.5 | 178.7 | 65.5 | 10.4 | 3.3 | 2.1 | 1.4 | 0.6 | 0.3 | 0.1 | 0.0 | |
| | MRL | 258 | 128 | 53 | 13 | 6 | 3 | 2 | 1 | 1 | 1 | 1 | |
| $gamma(5,1)$ | ARL | 371.8 | 193.4 | 79.8 | 16.9 | 6.1 | 3.4 | 2.2 | 1.3 | 1.0 | 1.0 | 1.0 | **10.0** |
| | SDRL | 368.1 | 190 | 74.0 | 11.7 | 3.5 | 2.1 | 1.4 | 0.6 | 0.2 | 0.1 | 0.0 | |
| | MRL | 255.5 | 136 | 58 | 14 | 6 | 3 | 2 | 1 | 1 | 1 | 1 | |

control chart designed by Yang et al. [26], the CUSUM sign control chart developed by Yang and Cheng [59], and the mixed EWMA-CUSUM sign control chart suggested by Abbasi et al. [60]. The comparison is based on $ARL_1$, $SDRL_1$, and $MRL_1$ values for a range of shifts in the process proportion, i.e., $\Delta = |p_0 - p_1|$. Moreover, considering a range of shifts in the process mean, i.e., $\mu_1 = \mu_0 + \delta\sigma$, the OOC RL profile comparison is made between the NPEWMA-MA sign chart and some existing nonparametric control charts for various symmetric and skewed distributions. For the overall performance measure, we have also computed the AEQL and RMI of the proposed and existing control charts.

For a rational RL profile comparison among the suggested and existing control charts, we fix sample size $n = 10$ and $ARL_0 \cong 370$, and accordingly the design parameters of the control charts under consideration are obtained. The RL profile of the NPEWMA-MA control chart using parameters $\eta = 0.05$, $w = 5$, and the existing arcsine MA sign control chart with $w = 5$, the EWMA sign control chart using $\eta = 0.05$, the CUSUM sign control chart with $K = 0.50$, $H = 10.60$, and the mixed EWMA-CUSUM sign control chart using $k = 0.50$, $h = 44.95$ are computed and presented in Table 8. Here, we have used arcsine transformation only for the

**Table 8. The RL profile of the MA sign, EWMA sign, CUSUM sign, mixed EWMA-CUSUM, and the EWMA-MA sign control charts for $n = 10$ at $ARL_0 \cong 370$.**

| $p_1$ | MA arcsine with $w = 5$ | | | EWMA sign with $\lambda = 0.05$ | | | CUSUM sign with $K = 0.50$ and $H = 10.60$ | | | Mixed EWMA-CUSUM sign with $k = 0.50$ and $h = 44.95$ | | | EWMA-MA sign with $\eta = 0.05$, $w = 5$ | | |
|---|---|---|---|---|---|---|---|---|---|---|---|---|---|---|---|
| | ARL | SDRL | MRL | ARL | SDRL | MRL | ARL | SDRL | MRL | ARL | SDRL | MRL | ARL | SDRL | MRL |
| 0.05 | 1.4 | 0.5 | 1 | 3.4 | 0.5 | 3 | 3.2 | 0.4 | 3 | 10.0 | 0.4 | 10 | **1.1** | 0.3 | 1 |
| 0.10 | 1.8 | 0.7 | 2 | 3.9 | 0.5 | 4 | 3.6 | 0.6 | 4 | 10.8 | 0.7 | 11 | **1.3** | 0.7 | 1 |
| 0.15 | 2.2 | 0.9 | 2 | 4.4 | 0.8 | 4 | 4.1 | 0.8 | 4 | 11.9 | 0.9 | 12 | **1.8** | 1.1 | 1 |
| 0.20 | 2.9 | 1.4 | 3 | 5.2 | 1.1 | 5 | 4.9 | 1.1 | 5 | 13.2 | 1.3 | 13 | **2.5** | 1.7 | 2 |
| 0.25 | 4.0 | 2.4 | 3 | 6.3 | 1.6 | 6 | 6.0 | 1.6 | 6 | 15.1 | 1.7 | 15 | **3.7** | 2.4 | 3 |
| 0.30 | 6.6 | 4.8 | 5 | 8.1 | 2.4 | 8 | 7.9 | 2.6 | 7 | 17.7 | 2.5 | 17 | **5.4** | 3.3 | 5 |
| 0.35 | 13.2 | 11.4 | 10 | 11.4 | 4.3 | 11 | 11.4 | 4.7 | 10 | 22.0 | 4.0 | 21 | **8.5** | 5.1 | 8 |
| 0.40 | 34.4 | 32.6 | 24 | 19.2 | 9.6 | 17 | 20.2 | 11.3 | 18 | 30.2 | 7.6 | 29 | **15.6** | 10.2 | 14 |
| 0.45 | 124.1 | 120.1 | 87 | 51.7 | 37.0 | 42 | 63.9 | 52.1 | 48 | 57.2 | 26.0 | 51 | **46.8** | 37.8 | 37 |
| 0.46 | 169.0 | 168.0 | 117 | 71.8 | 56.8 | 56 | 93.1 | 79.7 | 70 | 72.4 | 38.2 | 63 | **65.7** | 58.0 | 50 |
| 0.47 | 225.8 | 227.7 | 155 | 107.8 | 92.2 | 80 | 136.5 | 122.8 | 99 | 98.4 | 62.5 | 81 | **98.3** | 84.0 | 68 |
| 0.50 | 371.2 | 372.0 | 258 | 372.0 | 359.9 | 259 | 370.7 | 357.8 | 261 | 369.9 | 323.3 | 275 | 371.7 | 365.3 | 258 |
| 0.53 | 224.4 | 222.0 | 156 | 106.7 | 89.7 | 80 | 137.8 | 125.0 | 99 | 99.3 | 62.2 | 82 | **97.1** | 87.8 | 71 |
| 0.54 | 170.7 | 169.8 | 119 | 71.9 | 56.1 | 56 | 90.6 | 78.3 | 67 | 72.1 | 38.1 | 62 | **65.1** | 56.9 | 53 |
| 0.55 | 126.5 | 124.0 | 89 | 51.8 | 37.6 | 42 | 62.9 | 51.9 | 47 | 57.1 | 25.8 | 51 | **46.2** | 38.0 | 37 |
| 0.60 | 34.5 | 32.7 | 24 | 19.0 | 9.3 | 17 | 20.3 | 11.3 | 18 | 30.3 | 7.6 | 29 | **15.8** | 10.5 | 14 |
| 0.65 | 13.0 | 11.2 | 10 | 11.4 | 4.2 | 11 | 11.4 | 4.7 | 10 | 22.0 | 4.0 | 21 | **8.5** | 5.2 | 8 |
| 0.70 | 6.6 | 4.8 | 5 | 8.1 | 2.4 | 8 | 7.9 | 2.6 | 8 | 17.7 | 2.5 | 17 | **5.4** | 3.3 | 5 |
| 0.75 | 4.1 | 2.5 | 3 | 6.3 | 1.6 | 6 | 6.0 | 1.6 | 6 | 15.1 | 1.7 | 15 | **3.7** | 2.4 | 3 |
| 0.80 | 2.9 | 1.4 | 3 | 5.2 | 1.1 | 5 | 4.9 | 1.1 | 5 | 13.2 | 1.3 | 13 | **2.5** | 1.7 | 2 |
| 0.85 | 2.2 | 0.9 | 2 | 4.4 | 0.8 | 4 | 4.2 | 0.8 | 4 | 11.9 | 0.9 | 12 | **1.8** | 1.1 | 1 |
| 0.90 | 1.8 | 0.7 | 2 | 3.9 | 0.5 | 4 | 3.6 | 0.6 | 4 | 10.8 | 0.7 | 11 | **1.3** | 0.6 | 1 |
| 0.95 | 1.4 | 0.5 | 1 | 3.5 | 0.5 | 3 | 3.2 | 0.4 | 3 | 10.0 | 0.4 | 10 | **1.1** | 0.3 | 1 |

MA sign chart and the remaining charts are evaluated without arcsine transformation in Table 8. The reason for using arcsine transformation is that the existing MA sign chart without arcsine transformation does not achieve the desired $ARL_0$. The results shown in Table 8 indicate that the proposed control chart is more effective in detecting small to moderate shifts than the arcsine MA control chart. In comparison to the EWMA and CUSUM sign control charts, the NPEWMA-MA sign control chart performs much better for a range of shifts considered in this study. Moreover, it is noticed that the suggested NPEWMA-MA control chart performs significantly better for moderate to larger shifts as compared to the mixed EWMA-CUSUM sign chart and performs marginally better for small changes in the process proportion.

Table 9 presents the OOC RL properties of the MA sign, EWMA sign, CUSUM sign, EWMA-CUSUM sign, and NPEWMA-MA sign control charts using arcsine transformation for various symmetric and asymmetric distributions at $n = 10$ and $ARL_0 \cong 370$. The first row comprises of $ARL_1$s with $SDRL_1$s in parentheses, while $MRL_1$s are in the second row. It is observed that the OOC RL profile declines quickly with the increase in shift size. In addition, the lowest $ARL_1$, $SDRL_1$, and $MRL_1$ values for specific shifts and the smallest $AEQL$ and $RMI$ values for a range of shifts show that the proposed NPEWMA-MA sign control chart dominates its rivals irrespective of the type of distribution.

## 5. Practical example

For the empirical application of the proposed NPEWMA-MA sign control chart, a dataset from a combined cycle power plant (CCPP) was originally collected by Tüfekci [61]. The data

**Table 9. The OOC RL characteristics (the first row contains $ARL_1$ values with $SDRL_1$ values in parenthesis, while $MRL_1$ values are in the second row) of the MA sign, EWMA sign, CUSUM sign, mixed EWMA-CUSUM, and the EWMA-MA sign control charts using arcsine transformation for various continuous distributions at $n = 10$ and $ARL_0 \cong 370$.**

| Control Chart | δ | | | | | | | | | AEQL | RMI |
|---|---|---|---|---|---|---|---|---|---|---|---|
| | 0.1 | 0.25 | 0.50 | 0.75 | 1.00 | 1.50 | 2.00 | 2.5 | 3.0 | | |
| **Normal distribution i.e. N(0,1)** | | | | | | | | | | | |
| MA Sign | 171.7(169.5) | 35.8(34.4) | 7.3(5.5) | 3.4(1.8) | 2.3(1.0) | 1.5(0.6) | 1.2(0.4) | 1.1(0.2) | 1.0(0.1) | 11.74 | 0.45 |
| | 119 | 25 | 6 | 3 | 2 | 2 | 1 | 1 | 1 | | |
| EWMA Sign | 73.2(57.8) | 19.5(9.8) | 8.3(2.8) | 5.4(1.5) | 4.1(1.0) | 2.9(0.6) | 2.4(0.5) | 2.1(0.3) | 2(0.2) | 20.1 | 0.84 |
| | 57 | 17 | 8 | 5 | 4 | 3 | 2 | 2 | 2 | | |
| CUSUM sign | 92.8(80.9) | 20.8(11.8) | 8.3(2.8) | 5.4(1.3) | 4.3(0.8) | 3.3(0.5) | 3.0(0.2) | 3.0 (0.0) | 3.0 (0.0) | 26.5 | 1.21 |
| | 68 | 18 | 8 | 5 | 4 | 3 | 3 | 3 | 3 | | |
| EWMA-CUSUM sign | 74.2(39.2) | 31.2(8.0) | 18.5(2.9) | 13.9(1.8) | 11.4(1.3) | 8.9(0.8) | 7.8 (0.6) | 7.3(0.5) | 7.1(0.3) | 64.6 | 4.15 |
| | 64 | 30 | 18 | 14 | 11 | 9 | 8 | 7 | 7 | | |
| EWMA-MA sign | **64.1(55.7)** | **15.8(10.5)** | **5.9(3.6)** | **3.1(2.1)** | **1.9(1.2)** | **1.2(0.4)** | **1.0(0.1)** | **1.0(0.0)** | **1.0(0.0)** | **9.9** | **0** |
| | **49** | **14** | **6** | **3** | **1** | **1** | **1** | **1** | **1** | | |
| **Student's t-distribution with df = 5** | | | | | | | | | | | |
| MA Sign | 132.9(131.2) | 22.5(20.5) | 5.0(3.3) | 2.7(1.2) | 2.0(0.8) | 1.5(0.5) | 1.2(0.4) | 1.1(0.3) | 1.1(0.2) | 11.19 | 0.46 |
| | 93 | 16 | 4 | 3 | 2 | 1 | 1 | 1 | 1 | | |
| EWMA Sign | 54.7(40.4) | 15.1(6.8) | 6.8(2.1) | 4.6(1.2) | 3.6(0.9) | 2.8(0.6) | 2.4(0.5) | 2.2(0.4) | 2.1(0.3) | 19.9 | 0.96 |
| | 44 | 14 | 7 | 5 | 3 | 3 | 2 | 2 | 2 | | |
| CUSUM sign | 65.3(54.0) | 15.5(7.6) | 6.8(2.0) | 4.7(1.0) | 3.9(0.7) | 3.2(0.4) | 3.0(0.2) | 3.0(0.1) | 3.0(0.0) | 25.8 | 1.31 |
| | 49 | 14 | 6 | 4 | 4 | 3 | 3 | 3 | 3 | | |
| EWMA-CUSUM sign | 59.7(26.9) | 26.8(5.9) | 16.2(2.3) | 12.4(1.5) | 10.5(1.1) | 8.7(0.7) | 7.9(0.6) | 7.5(0.5) | 7.3(0.5) | 64.7 | 4.58 |
| | 54 | 26 | 16 | 12 | 10 | 9 | 8 | 8 | 7 | | |
| EWMA-MA sign | **47.2(42.1)** | **11.9(7.5)** | **4.4(2.7)** | **2.3(1.5)** | **1.5(0.9)** | **1.1(0.4)** | **1.0(0.2)** | **1.0(0.1)** | **1.0(0.0)** | **9.3** | **0** |
| | **36** | **11** | **4** | **2** | **1** | **1** | **1** | **1** | **1** | | |
| **Laplace distribution i.e. $Laplace\left(0, \frac{1}{\sqrt{2}}\right)$** | | | | | | | | | | | |
| MA Sign | 80.9(78.7) | 13.1(11.6) | 3.9(2.3) | 2.5(1.1) | 1.9(0.8) | 1.5 (0.6) | 1.3 (0.4) | 1.1 (0.3) | 1.1(0.3) | 10.78 | 0.41 |
| | 57 | 10 | 3 | 2 | 2 | 1 | 1 | 1 | 1 | | |
| EWMA Sign | 35.1 (22.7) | 11.4 (4.5) | 5.9 (1.7) | 4.3 (1.1) | 3.5 (0.8) | 2.8 (0.6) | 2.5 (0.5) | 2.3 (0.4) | 2.1 (0.3) | 20.0 | 0.99 |
| | 29 | 11 | 6 | 4 | 3 | 3 | 2 | 2 | 2 | | |
| CUSUM sign | 39.3(28.1) | 11.5(4.7) | 5.9(1.6) | 4.5(0.9) | 3.8(0.6) | 3.3(0.5) | 3.1(0.2) | 3.0(0.1) | 3.0(0.0) | 25.7 | 1.34 |
| | 32 | 11 | 6 | 4 | 4 | 3 | 3 | 3 | 3 | | |
| EWMA-CUSUM sign | 44.5(15.6) | 22.5(4.2) | 14.8(2.0) | 11.9(1.4) | 10.4(1.1) | 8.8(0.8) | 8.0 (0.6) | 7.6(0.5) | 7.4(0.5) | 65.0 | 4.76 |
| | 41 | 22 | 15 | 12 | 10 | 9 | 8 | 8 | 7 | | |
| EWMA-MA sign | **29.4(22.1)** | **8.6(5.3)** | **3.6(2.3)** | **2.1(1.4)** | **1.5(0.8)** | **1.1(0.4)** | **1.0(0.2)** | **1.0(0.1)** | **1.0(0.0)** | **9.0** | **0** |
| | **24** | **8** | **3** | **2** | **1** | **1** | **1** | **1** | **1** | | |
| **Logistic distribution i.e. $LG\left(0, \frac{\sqrt{3}}{\pi}\right)$** | | | | | | | | | | | |
| MA Sign | 149.4(148.0) | 26.8(24.5) | 5.7(4.0) | 2.9(1.4) | 2.1(0.8) | 1.5 (0.6) | 1.2 (0.4) | 1.1 (0.3) | 1.0(0.2) | 11.16 | 0.43 |
| | 102 | 19 | 5 | 3 | 2 | 1 | 1 | 1 | 1 | | |
| EWMA Sign | 60.4(45.8) | 16.6(7.9) | 7.3(2.4) | 4.9(1.3) | 3.8(0.9) | 2.8(0.6) | 2.4(0.5) | 2.2(0.4) | 2.1(0.3) | 20.2 | 0.91 |
| | 48 | 15 | 7 | 5 | 4 | 3 | 2 | 2 | 2 | | |
| CUSUM sign | 74.5(63.1) | 17.1(8.9) | 7.3(2.3) | 5.0(1.2) | 4.1(0.7) | 3.3(0.5) | 3.0(0.2) | 3.0(0.1) | 3.0(0.0) | 26.1 | 1.27 |
| | 55 | 15 | 7 | 5 | 4 | 3 | 3 | 3 | 3 | | |
| EWMA-CUSUM sign | 64.7(30.8) | 28.3(6.5) | 17.1(2.5) | 13.0(1.6) | 10.9(1.2) | 8.8(0.8) | 7.9(0.6) | 7.5(0.5) | 7.2(0.4) | 64.8 | 4.43 |
| | 57 | 27 | 17 | 13 | 11 | 9 | 8 | 7 | 7 | | |
| EWMA-MA sign | **55.5(43.6)** | **13.4(8.5)** | **4.8(3.0)** | **2.6(1.7)** | **1.7(1.0)** | **1.1(0.4)** | **1.0(0.2)** | **1.0(0.1)** | **1.0(0.0)** | **9.5** | **0** |
| | **45** | **12** | **5** | **2** | **1** | **1** | **1** | **1** | **1** | | |

*(Continued)*

**Table 9.** (Continued)

| Control Chart | δ | | | | | | | | | AEQL | RMI |
|---|---|---|---|---|---|---|---|---|---|---|---|
| | 0.1 | 0.25 | 0.50 | 0.75 | 1.00 | 1.50 | 2.00 | 2.5 | 3.0 | | |
| **Contaminated Normal distribution with a 10% contamination proportion** | | | | | | | | | | | |
| MA Sign | 152.2(151.5) | 29.1(27.0) | 6.1(4.4) | 3.1(1.6) | 2.2(0.9) | 1.5(0.6) | 1.2(0.4) | 1.1(0.2) | 1.0(0.1) | 11.33 | 0.46 |
| | 106 | 21 | 5 | 3 | 2 | 1 | 1 | 1 | 1 | | |
| EWMA Sign | 64.0(49.6) | 17.4(8.4) | 7.6(2.5) | 5.0(1.4) | 3.9(1) | 2.8(0.6) | 2.4(0.5) | 2.1(0.3) | 2(0.2) | 19.8 | 0.89 |
| | 50 | 16 | 7 | 5 | 4 | 3 | 2 | 2 | 2 | | |
| CUSUM sign | 78.3(66.7) | 18.1(9.7) | 7.6(2.5) | 5.1(1.2) | 4.1(0.7) | 3.3(0.5) | 3.0(0.2) | 3.0(0.0) | 3.0(0.0) | 26.2 | 1.27 |
| | 58 | 16 | 7 | 5 | 4 | 3 | 3 | 3 | 3 | | |
| EWMA-CUSUM sign | 67.5(33.1) | 29.2(6.9) | 17.4(2.6) | 13.3(1.7) | 11.0(1.2) | 8.8(0.8) | 7.8(0.6) | 7.3(0.5) | 7.1(0.2) | 64.1 | 4.33 |
| | 59 | 28 | 17 | 13 | 11 | 9 | 8 | 7 | 7 | | |
| EWMA-MA sign | **54.5(47.2)** | **14.0(9.0)** | **5.1(3.2)** | **2.7(1.8)** | **1.8(1.1)** | **1.1(0.4)** | **1.0(0.1)** | **1.0(0.0)** | **1.0(0.0)** | **9.6** | **0** |
| | **44** | **13** | **5** | **2** | **1** | **1** | **1** | **1** | **1** | | |
| **Gamma distribution, i.e., Gamma(3,1)** | | | | | | | | | | | |
| MA Sign | 160.3(159.0) | 32.5(30.5) | 7.1(5.3) | 3.4(1.9) | 2.4(1.0) | 1.6(0.6) | 1.3(0.5) | 1.1(0.3) | 1.0(0.2) | 11.9 | 0.46 |
| | 113 | 23 | 5 | 3 | 2 | 2 | 1 | 1 | 1 | | |
| EWMA Sign | 68.1(53.4) | 18.5(9.2) | 8.2(2.8) | 5.4(1.5) | 4.2(1.1) | 3(0.7) | 2.5(0.6) | 2.2(0.4) | 2.1(0.3) | 20.9 | 0.88 |
| | 53 | 17 | 8 | 5 | 4 | 3 | 3 | 2 | 2 | | |
| CUSUM sign | 84.1(72.7) | 19.4(10.6) | 8.2(2.7) | 5.5(1.4) | 4.3(0.9) | 3.4(0.5) | 3.1(0.3) | 3.0(0.1) | 3.0(0.0) | 26.6 | 1.22 |
| | 62 | 17 | 8 | 5 | 4 | 3 | 3 | 3 | 3 | | |
| EWMA-CUSUM sign | 70.4(35.7) | 30.3(7.4) | 18.3(2.9) | 14.0(1.8) | 11.7(1.3) | 9.2(0.9) | 8.2(0.6) | 7.6(0.5) | 7.3(0.4) | 66.7 | 4.29 |
| | 61 | 29 | 18 | 14 | 12 | 9 | 8 | 8 | 7 | | |
| EWMA-MA sign | **59.9(49.8)** | **15.1(10.0)** | **5.6(3.5)** | **3.1(2.1)** | **2.0(1.3)** | **1.2(0.5)** | **1.0(0.2)** | **1.0(0.1)** | **1.0(0.0)** | **9.9** | **0** |
| | **47** | **13** | **6** | **3** | **2** | **1** | **1** | **1** | **1** | | |
| **Weibull distribution, i.e., Weibull(2,1)** | | | | | | | | | | | |
| MA Sign | 198.4(197.0) | 53.8(52.3) | 14.0(12.0) | 6.6(4.9) | 4.4(2.7) | 2.8(1.3) | 2.2(0.9) | 2.0(0.8) | 1.8(0.7) | 20.95 | 0.54 |
| | 137 | 37 | 10 | 5 | 4 | 3 | 2 | 2 | 2 | | |
| EWMA Sign | 89.7 (74.1) | 25.5 (14.7) | 11.6 (4.6) | 7.9 (2.6) | 6.3 (1.9) | 4.7 (1.3) | 4.0 (1.0) | 3.5 (0.9) | 3.3 (0.8) | 32.5 | 0.74 |
| | 68 | 22 | 11 | 8 | 6 | 5 | 4 | 3 | 3 | | |
| CUSUM sign | 113.2(102.9) | 27.8(17.5) | 11.7(4.9) | 7.9(2.6) | 6.3(1.8) | 4.8(1.1) | 4.2(0.8) | 3.8(0.6) | 3.6(0.6) | 34.6 | 0.85 |
| | 82 | 23 | 11 | 8 | 6 | 5 | 4 | 4 | 4 | | |
| EWMA-CUSUM sign | 86.6(48.3) | 36.8(11.0) | 22.8(4.3) | 18.0(2.8) | 15.4(2.1) | 12.7(1.5) | 11.2(1.3) | 10.4(1.1) | 9.7(1.0) | 89.7 | 3.26 |
| | 74 | 35 | 22 | 18 | 15 | 13 | 11 | 10 | 10 | | |
| EWMA-MA sign | **82.7(76.2)** | **21.6(15.5)** | **8.8(5.3)** | **5.4(3.4)** | **3.9(2.5)** | **2.5(1.6)** | **1.8(1.1)** | **1.5(0.8)** | **1.3(0.7)** | **15.6** | **0** |
| | **60** | **18** | **8** | **5** | **3** | **2** | **1** | **1** | **1** | | |

consists of 9568 observations collected over a period of 6 years from 2006 to 2011. The combined cycle power plant produces electrical power through the mixture of steam and gas turbines being combined in the cycle. The four input variables of the dataset are relative humidity (RH), ambient pressure (AP), ambient temperature (AT), and exhaust vacuum (EV) which can affect the CCPP's ability to generate electricity. Here, we use ambient temperature (AT) as a study variable that might impact the gas turbine's performance (details can be found in Tüfekci [61]). The mean value of AT is 19.65 and the standard deviation is 7.452 under the IC process. The value of the coefficient of skewness is -0.136 which indicates that the distribution of AT is negatively skewed.

For assessing the independence and randomness of observations, the correlation between successive observations (r = −0.006) and the run test for randomness (Test Statistics = 0.51762, p−value = 0.6047) of AT are computed, respectively. The results indicate that the observations

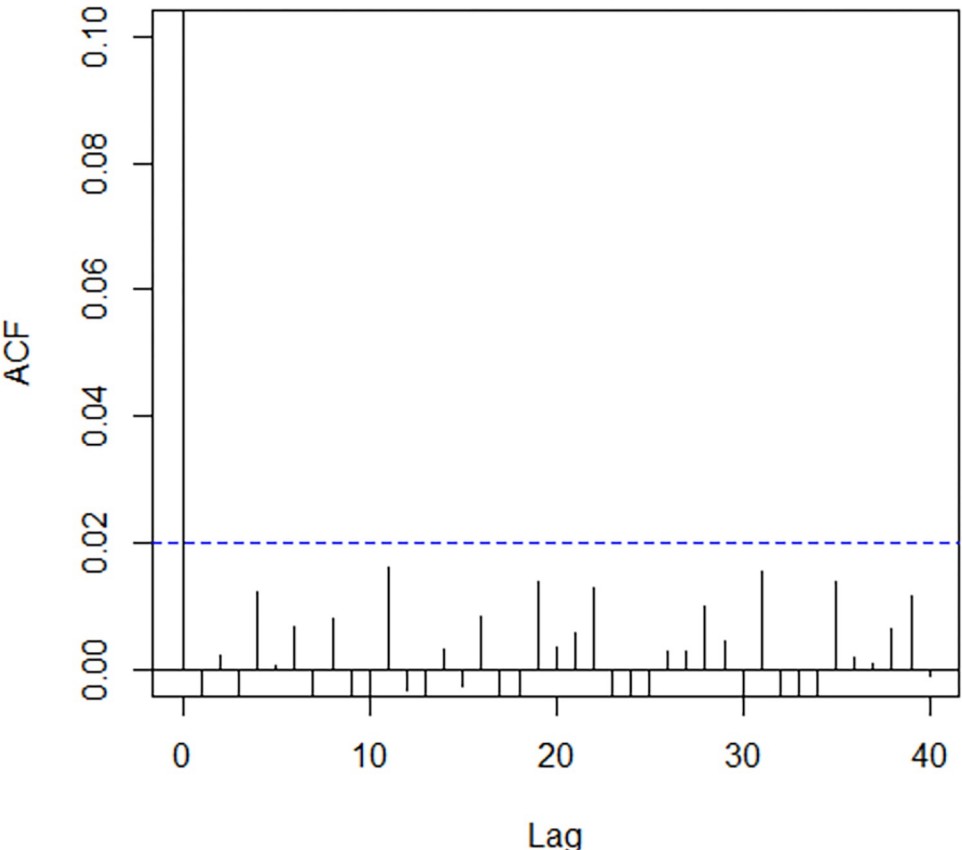

**Fig 1. ACF plot for the ambient temperature (AT) data.**

are independent and random. Moreover, the autocorrelation function (ACF) in R-language is used to assess the autocorrelation at various lags of AT data. It can be observed from Fig 1 that the autocorrelation coefficients at different lags are very close to zero indicating that the AT observations are serially independent. To determine the normality of AT, the Anderson-Darling (A = 85.528, p−value = 0.000) and the Shapiro-Wilk (W = 0.97254, p−value = 0.000) tests are used. These findings show that the data are not normally distributed, as the p-values of both tests are sufficiently small. For this type of quality characteristic, nonparametric control charts are robust alternatives for monitoring process parameters.

We take 50 samples, each of size 10, from the AT dataset. The first 20 samples of size 10 each are drawn from the IC process state, with a median of 19.96, while the next 30 samples are obtained by shifting the process average by 0.25σ. To identify changes in process location, suggested and existing control charts are constructed by setting $ARL_0 \cong 370$. The nonparametric arcsine MA sign control chart is constructed using L = 3.10 and w = 5, the nonparametric arcsine EWMA sign control chart is computed using L = 2.675 and η = 0.05, the CUSUM sign control chart is calculated using charting parameters K = 0.50 and H = 10.60, the arcsine mixed EWMA-CUSUM sign chart is constructed using k = 0.50 and h = 51.28, and the proposed arcsine NPEW-MA-MA chart is established using L = 2.305, w = 5, and η = 0.05. Figs 2 through 6, respectively, show the plotting statistics for these control charts against their corresponding control limits.

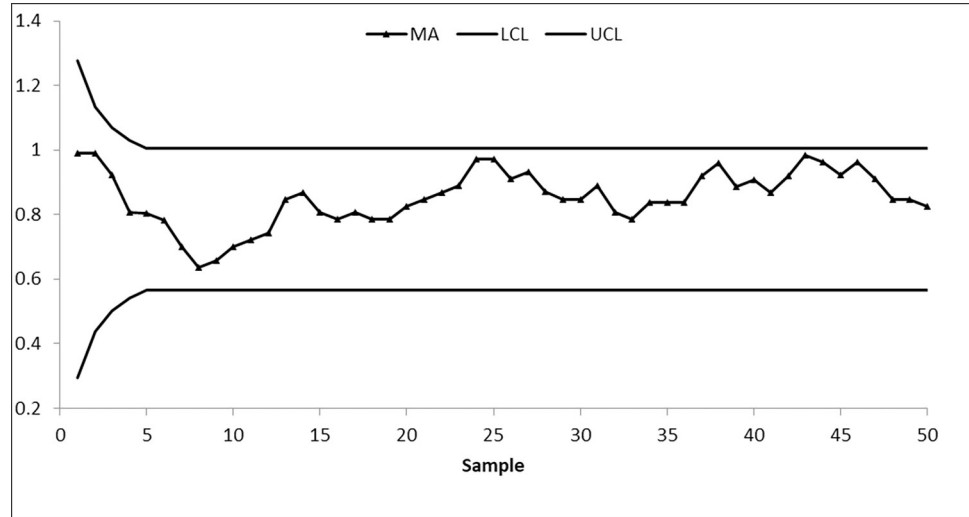

**Fig 2. Nonparametric MA arcsine control chart at w = 5 and 〖ARL〗_0≅370.**

From Figs 2 and 5, it is evident that the existing arcsine MA and arcsine mixed EWMA-CUSUM sign control charts do not detect a process shift, indicating that the process is IC. Whereas, Figs 3 and 4 depict that both existing arcsine EWMA and CUSUM sign control charts detect the shift at sample 42, meaning that on average 22 samples are required to detect this shift. The proposed chart displayed in Fig 6 detects the process shift at sample point 38, i.e., on average 18 samples are needed to identify this shift in the process. These findings further confirm that the EWMA-MA sign control chart has significantly better ability to identify shifts compared to other existing charts considered in this study.

## 6. Conclusion

In this study, we propose a mixed NPEWMA-MA sign control chart that integrates the corrected variance of EWMA-MA statistics to monitor changes in the location parameter. The

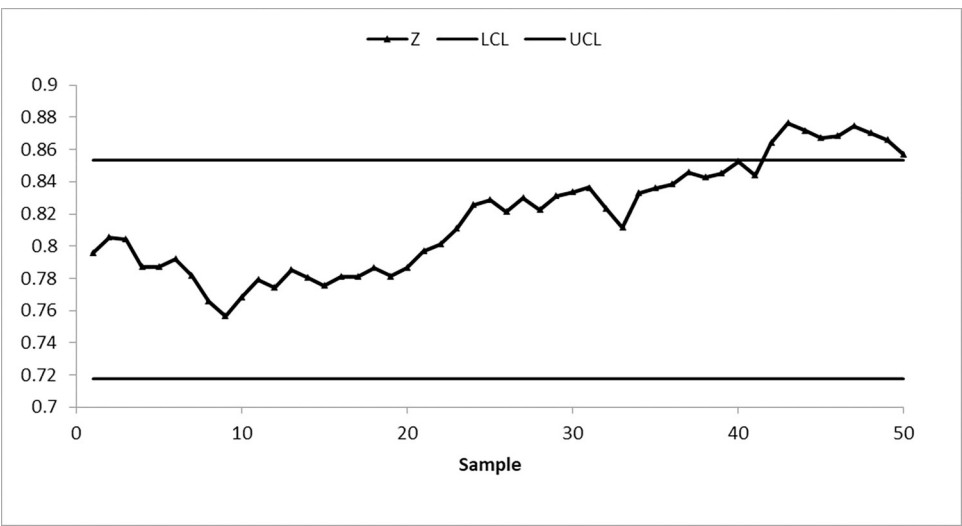

**Fig 3. Nonparametric EWMA arcsine control chart at η = 0.05 and 〖ARL〗_0≅370.**

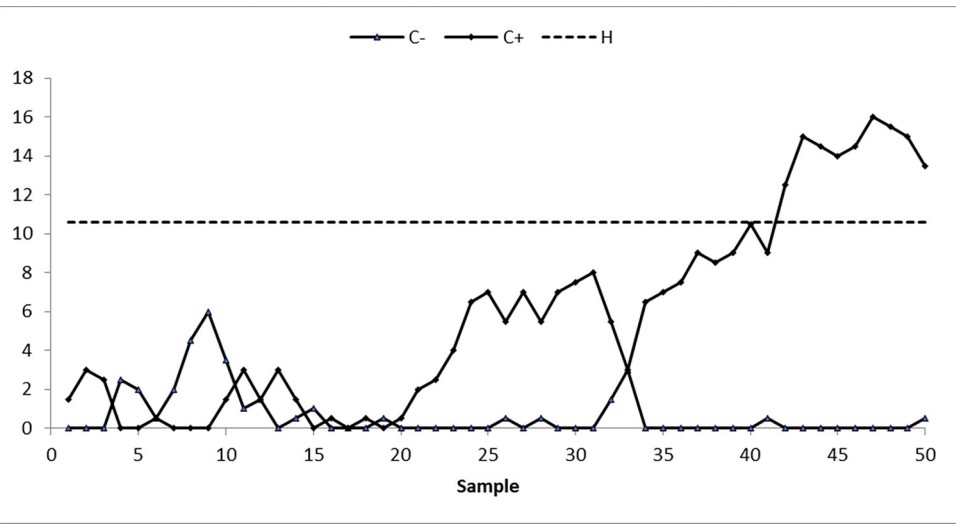

**Fig 4. Nonparametric CUSUM sign control chart at K = 0.50, H = 10.60, and** $\llbracket ARL \rrbracket\_0 \cong 370$.

charting statistic combines the MA statistic with the EWMA statistic. Monte Carlo simulation is employed to determine the RL profile. The performance of the proposed chart is evaluated under different RL features such as ARL, SDRL, and MRL. Aditionally, AEQL and RMI are calculated as overall performance measures. Based on these assessments, the proposed NPEW-MA-MA sign control chart demonstrates superior efficiency compared to the existing control charts considered in this study. Furthermore, the performance of the proposed chart is evaluated under various symmetrical and skewed distributions, highlighting its robustness and enhanced capability to detect shifts in the process location. A practical application is presented to illustrate the effectiveness of the proposed chart in promptly identifying process shifts. Therefore, we recommend using the proposed NPEWMA-MA sign control chart, with or without arcsine transformation, especially when dealing with non-normal or unknown distributions in quality management practices.

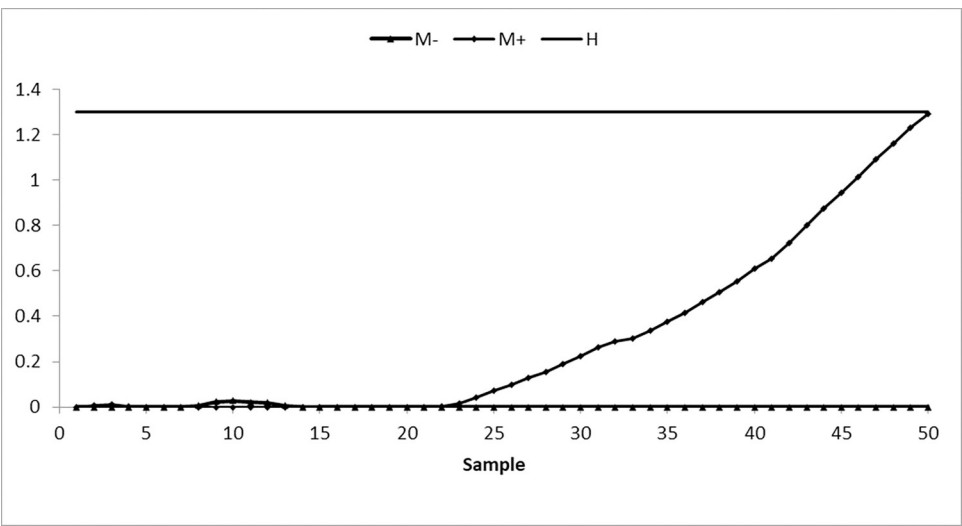

**Fig 5. Nonparametric mixed EWMA-CUSUM sign control chart at λ = 0.05, k = 0.50, h = 51.28, and** $\llbracket ARL \rrbracket\_0 \cong 370$.

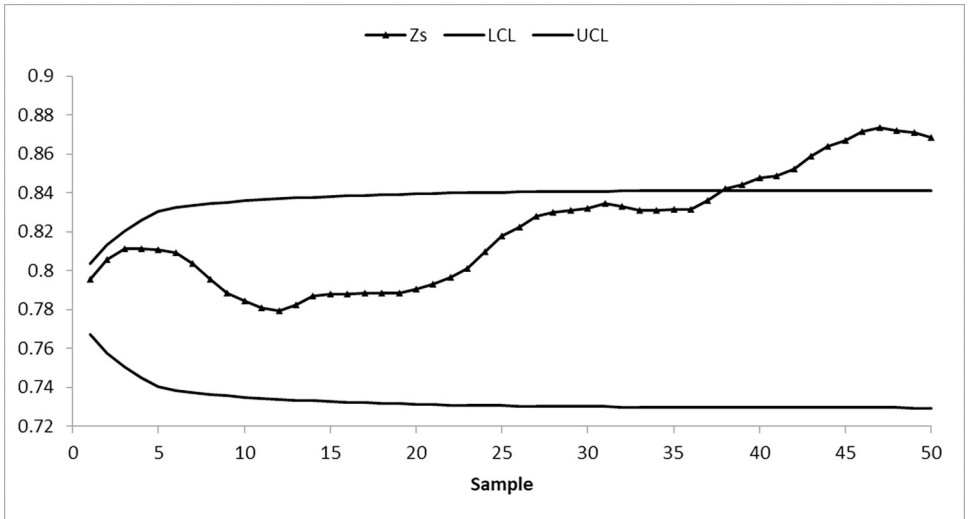

**Fig 6. Nonparametric EWMA-MA arcsine control chart at η = 0.05, w = 5, and ⟦ARL⟧ _0≅370.**

## Author Contributions

**Conceptualization:** Muhammad Ali Raza, Farah Tariq.

**Data curation:** Muhammad Ali Raza, Farah Tariq.

**Formal analysis:** Gideon Mensah Engmann, Ali M. Mahnashi.

**Investigation:** Abdullah A. Zaagan, Mutum Zico Meetei.

**Methodology:** Muhammad Ali Raza, Mutum Zico Meetei.

**Project administration:** Gideon Mensah Engmann.

**Resources:** Farah Tariq, Abdullah A. Zaagan, Gideon Mensah Engmann, Ali M. Mahnashi.

**Software:** Abdullah A. Zaagan, Gideon Mensah Engmann.

**Supervision:** Ali M. Mahnashi.

**Validation:** Muhammad Ali Raza, Gideon Mensah Engmann, Ali M. Mahnashi, Mutum Zico Meetei.

**Visualization:** Abdullah A. Zaagan.

**Writing – original draft:** Muhammad Ali Raza, Farah Tariq, Ali M. Mahnashi.

**Writing – review & editing:** Muhammad Ali Raza, Abdullah A. Zaagan, Mutum Zico Meetei.

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
