## [Decision Letter · Decision Letter 0]

12 Jun 2024

PONE-D-24-18713A Nonparametric Mixed Exponentially Weighted Moving Average- Moving Average Control Chart with an Application to Gas TurbinesPLOS ONE

Dear Dr. Meetei,

Thank you for submitting your manuscript to PLOS ONE. After careful consideration, we feel that it has merit but does not fully meet PLOS ONE’s publication criteria as it currently stands. Therefore, we invite you to submit a revised version of the manuscript that addresses the points raised during the review process.

Dear Authors, We have now recieved review reports about your manuscript. Both the referees are primarily in favour of article but with some minor concerns. You are invited to submit a revised version of manuscript by addressing the issues raised by referees in their comments.

We look forward to receiving your revised manuscript.

Kind regards,

Sajjad Haider Bhatti, Ph.D.

Academic Editor

PLOS ONE

Journal Requirements:

   "This research was funded by the Deputyship for Research and Innovation, Ministry of

Education in Saudi Arabia ISP-2024."

Additional Editor Comments:

Dear Authors,

We have now recieved review reports about your manuscript. Both the referees are primarily in favour of article but with some minor concerns.

You are invited to submit a revised version of manuscript by addressing the issues raised by referees in their comments.

Reviewers' comments:

Reviewer's Responses to Questions

**Comments to the Author**

1. Is the manuscript technically sound, and do the data support the conclusions?

Reviewer #1: Yes

Reviewer #2: Yes

2. Has the statistical analysis been performed appropriately and rigorously? 

Reviewer #1: Yes

Reviewer #2: Yes

3. Have the authors made all data underlying the findings in their manuscript fully available?

Reviewer #1: Yes

Reviewer #2: Yes

4. Is the manuscript presented in an intelligible fashion and written in standard English?

Reviewer #1: Yes

Reviewer #2: No

5. Review Comments to the Author

Reviewer #1: After critical review, it is observed that authors developed a Nonparametric Mixed Exponentially Weighted Moving Average-Moving Average Control Chart. The manuscript is mathematically and technically very sound and has vide applicability in industry. Though, I have some observations:

1. The abstract should be re-written in more concise way including objective, methodology and findings.

2. The manuscript is very lengthy. To increase the readability, reduce the length of manuscript if possible.

3. All abbreviation should be defined as and when first time used.

4. The experimental environment should be included in text.

Reviewer #2: Reviewer’s Comments

Title: A Nonparametric Mixed Exponentially Weighted Moving Average-Moving Average Control Chart with an Application to Gas Turbines

Manuscript ID: PONE-D-24-18713

The authors developed a nonparametric mixed exponentially weighted moving average-moving average sign control chart (NPEWMA-MA) by integrating the moving average statistic into the exponentially weighted moving average statistic. The charting strategy was previously discussed by Sukparungsee et al. [1] in a parametric environment, assuming the successive moving averages as constant. This study, however, explores the nonparametric structure of this charting strategy and also considers covariance terms in the variance expression, emphasizing that the successive MAs are not independent, as they utilize information from the previous w-1 samples, which was lacking in the existing study by Sukparungsee et al. [1]. Simulation study and real-life example are provided for practical implementation and comparison purposes. The paper is well-written, but the following observations need to be addressed.

Comment 1. In the algorithm to obtain the run length profiles: It seems that steps 2 and 7 are contradictory. You have already set the parameters for a fixed value of ARL0, so why do you check if the desired ARL0 is attained?

Comment 2. Include the recently published paper [2] in Introduction Section, where the authors provided a comprehensive simulation study of zero state and steady state rung-length properties of mixed control charts.

Comment 3. Below Table 8, in the comparison of different charts, why were these chart parameters chosen? Please clarify this.

Comment 4. This paper focuses on scenarios where the normality assumption is not valid. However, in practice, the assumption of independence might also be invalid. The authors should, clarify this issue in revision.

Comment 5. The manuscript is generally readable, but there are few typos present. It should undergo careful proofreading. Additionally, expanded form of abbreviations used in the manuscript must be provided when they are first introduced for clarity and comprehension.

[1]. Sukparungsee, S., Areepong, Y. & Taboran, R. (2020) Exponentially weighted moving average—Moving average charts for monitoring the process mean. PLoS One. 15(2), e0228208.

[2]. Alevizakos, V., Chatterjee, K., & Koukouvinos, C. (2024). On the performance and comparison of various memory-type control charts. Communications in Statistics - Simulation and Computation, 1–21. https://doi.org/10.1080/03610918.2024.2310692

6. PLOS authors have the option to publish the peer review history of their article (what does this mean?). If published, this will include your full peer review and any attached files.

Reviewer #1: No

Reviewer #2: No

---

## [Author Response · Author response to Decision Letter 0]

29 Jun 2024

Response Letter

Manuscript ID: PONE-D-24-18713

Manuscript Title: A Nonparametric Mixed Exponentially Weighted Moving Average-Moving Average Control Chart with an Application to Gas Turbines

Respected Editor and Reviewers, 

Thank you very much for your valuable suggestions for the betterment of the manuscript. Following is the itemize response to the comments/suggestions. 

Reviewer 1

Comment 1: The abstract should be re-written in more concise way including objective, methodology and findings.

Response: We have revised the manuscript as suggested.

Comment 2: The manuscript is very lengthy. To increase the readability, reduce the length of manuscript if possible.

Response: We have incorporated the suggestion in the revised manuscript by reducing the interpretations of the results.

Comment 3: All abbreviation should be defined as and when first time used.

Response: The suggestion is incorporated in the revised manuscript.

Comment 4: The experimental environment should be included in text.

Response: The suggestion is incorporated in the revised manuscript.

Reviewer 2

Comment 1: In the algorithm to obtain the run length profiles: It seems that steps 2 and 7 are contradictory. You have already set the parameters for a fixed value of ARL0, so why do you check if the desired ARL0 is attained?

Response: Initially, We find the value of limit coefficient (L) for fixed values of other design parameters η and w in order to achieve the desired 〖ARL〗_0. If the desired 〖ARL〗_0 is not attained at that particular value of L, we revise the value of L to attain desired 〖ARL〗_0. This procedure is repeated until we get the desired 〖ARL〗_0. After that, we have obtained the IC and OOC run-length characteristics using the finalized value of L. 

Comment 2: Include the recently published paper [2] in Introduction Section, where the authors provided a comprehensive simulation study of zero state and steady state run-length properties of mixed control charts.

Response: The suggested paper has been cited in the revised manuscript.

Comment 3: Below Table 8, in the comparison of different charts, why were these chart parameters chosen? Please clarify this.

Response: For a rational comparison between the existing and proposed control charts in Table 8, we choose those values of design parameters of the existing and proposed control charts for which 〖ARL〗_0≅370. In Table 8 we compare the performance of the proposed control chart with the existing control charts on the basis of shift in process proportion. Whereas, in Table 9, we compare the performance of existing and proposed control charts on the basis of various symmetric and asymmetric distributions under various parameter(s) settings.

Comment 4: This paper focuses on scenarios where the normality assumption is not valid. However, in practice, the assumption of independence might also be invalid. The authors should, clarify this issue in revision.

Response: This proposed control chart initially assumes that the observations must be i.i.d. (independently and identically distributed), which is the basic assumption of the sign test as described by Yang et al. (2011) and many other authors who have utilized the sign or arc sign test in their charting structure. So, our charting structure is confined to unknown or non-normal but i.i.d. observations, as already mentioned in Section 2.

Comment 5: The manuscript is generally readable, but there are few typos present. It should undergo careful proofreading. Additionally, expanded form of abbreviations used in the manuscript must be provided when they are first introduced for clarity and comprehension.

Response: The suggestion is incorporated in the revised manuscript.

Yang, S.-F., Lin, J.-S. & Cheng, S. W. A new nonparametric EWMA sign control chart. Expert. Syst. Appl. 38(5), 6239-6243 (2011).

---

## [Editor Report · Decision Letter 1]

9 Jul 2024

A Nonparametric Mixed Exponentially Weighted Moving Average- Moving Average Control Chart with an Application to Gas Turbines

PONE-D-24-18713R1

Dear Dr. Meetei,

We’re pleased to inform you that your manuscript has been judged scientifically suitable for publication and will be formally accepted for publication once it meets all outstanding technical requirements.

Kind regards,

Sajjad Haider Bhatti, Ph.D.

Academic Editor

PLOS ONE

Additional Editor Comments (optional):

The Authors have revised the manuscript in light of the comments by reviewers.

The article now stands fit for publication in PLOS One, Therefore, I recommend it for production/publication.
---

## [Editor Report · Acceptance letter]

2 Aug 2024

PONE-D-24-18713R1 

PLOS ONE

Dear Dr. Meetei, 

I'm pleased to inform you that your manuscript has been deemed suitable for publication in PLOS ONE. Congratulations! Your manuscript is now being handed over to our production team.

Kind regards, 

on behalf of

Dr. Sajjad Haider Bhatti 

Academic Editor

PLOS ONE